# The *wtf4* meiotic driver utilizes controlled protein aggregation to generate selective cell death

**Nicole L Nuckolls**[1], **Anthony C Mok**[1,2], **Jeffrey J Lange**[1], **Kexi Yi**[1], **Tejbir S Kandola**[1,3], **Andrew M Hunn**[1], **Scott McCroskey**[1], **Julia L Snyder**[1], **María Angélica Bravo Núñez**[1], **Melainia McClain**[1], **Sean A McKinney**[1], **Christopher Wood**[1], **Randal Halfmann**[1,4], **Sarah E Zanders**[1,4]\*

[1]Stowers Institute for Medical Research, Kansas City, United States; [2]University of Missouri-Kansas City, Kansas City, United States; [3]Open University, Milton Keynes, United Kingdom; [4]Department of Molecular and Integrative Physiology, University of Kansas Medical Center, Kansas City, United States

**Abstract** Meiotic drivers are parasitic loci that force their own transmission into greater than half of the offspring of a heterozygote. Many drivers have been identified, but their molecular mechanisms are largely unknown. The *wtf4* gene is a meiotic driver in *Schizosaccharomyces pombe* that uses a poison-antidote mechanism to selectively kill meiotic products (spores) that do not inherit *wtf4*. Here, we show that the Wtf4 proteins can function outside of gametogenesis and in a distantly related species, *Saccharomyces cerevisiae*. The Wtf4$^{poison}$ protein forms dispersed, toxic aggregates. The Wtf4$^{antidote}$ can co-assemble with the Wtf4$^{poison}$ and promote its trafficking to vacuoles. We show that neutralization of the Wtf4$^{poison}$ requires both co-assembly with the Wtf4$^{antidote}$ and aggregate trafficking, as mutations that disrupt either of these processes result in cell death in the presence of the Wtf4 proteins. This work reveals that *wtf* parasites can exploit protein aggregate management pathways to selectively destroy spores.

\*For correspondence:
sez@stowers.org

## Introduction

Meiotic drivers are selfish DNA sequences that break the traditional rules of sexual reproduction. Whereas most alleles have a 50% chance of being transmitted into a given offspring, meiotic drivers can manipulate gametogenesis to bias their own transmission into most or even all of an individual's offspring (*Burt and Trivers, 2006*; *Lindholm et al., 2016*). This makes meiotic drive a powerful evolutionary force (*Sandler and Novitski, 1957*). Meiotic drivers are widespread in eukaryotes and the evolutionary pressures they exert are thought to shape major facets of gametogenesis, including recombination landscapes and chromosome structure (*Bravo Núñez et al., 2020b*; *Bravo Núñez et al., 2020a*; *Crow, 1991*; *Dyer et al., 2007*; *Larracuente and Presgraves, 2012*; *Schimenti, 2000*; *Pardo-Manuel de Villena and Sapienza, 2001*; *Hammer et al., 1989*; *Zanders et al., 2014*; *Grey et al., 2018*).

Harnessing and wielding the evolutionary power of meiotic drive has the potential to greatly benefit humanity. Engineered drive systems, known as 'gene drives,' are being developed to spread genetic traits in populations (*Lindholm et al., 2016*; *Burt, 2014*; *Gantz et al., 2015*; *Esvelt et al., 2014*; *Burt and Crisanti, 2018*). For example, gene drives could be used to spread disease-resistance alleles in crops. Alternatively, gene drives can be used to suppress human disease vectors, such as mosquitoes, or to limit their ability to transmit diseases (*Burt, 2014*; *Burt and Crisanti, 2018*; *Esvelt et al., 2014*; *Gantz et al., 2015*; *Lindholm et al., 2016*). While there are many challenges involved in designing effective gene drives, natural meiotic drivers could serve as useful

**eLife digest** Meiotic drivers are genes that break the normal rules of inheritance. Usually, a gene has a 50% chance of passing on to the next generation. Meiotic drivers force their way into the next generation by poisoning the gametes (the sex cells that combine to form a zygote) that do not carry them. Harnessing the power of genetic drivers could allow scientists to spread beneficial genes across populations.

One group of meiotic drivers found in fission yeast is called the 'with transposon fission yeast', or '*wtf*' gene family. The *wtf* drivers act during the production of spores, which are the fission yeast equivalent of sperm, and they encode both a poison that can destroy the spores and its antidote. The poison spreads through the sac holding the spores, and can affect all of them, while the antidote only protects the spores that make it. This means that the spores carrying the *wtf* genes survive, while the rest of the spores are killed. To understand whether it is possible to use the *wtf* meiotic drivers to spread other genes, perhaps outside of fission yeast, scientists must first establish exactly how the proteins coded for by genes behave.

To do this, Nuckolls et al. examined a member of the *wtf* family called *wtf4*. Attaching a fluorescent label to the poison and antidote proteins produced by *wtf4* made it possible to see what they do. This revealed that the poison clumps, forming toxic aggregates that damage yeast spores. The antidote works by mopping up these aggregates and moving them to the cell's main storage compartment, called the vacuole. Mutations that disrupted the ability of the antidote to interact with the poison or its ability to move the poison into storage stopped the antidote from working. Nuckolls et al. also showed that if genetic engineering was used to introduce *wtf4* into a distantly related species of budding yeast the effects of this meiotic driver were the same. This suggests that the *wtf* genes may be good candidates for future genetic engineering experiments.

Engineered systems known as 'gene drives' could spread beneficial genetic traits through populations. This could include disease-resistance genes in crops, or disease-preventing genes in mosquitoes. The *wtf* genes are small and work independently of other genes, making them promising candidates for this type of system. These experiments also suggest that the *wtf* genes could be useful for understanding why clumps of proteins are toxic to cells. Future work could explore why clumps of *wtf* poison kill spores, while clumps of poison plus antidote do not. This could aid research into human ailments caused by protein clumps, such as Huntington's or Alzheimer's disease.

models or components for these systems (*Burt, 2014*; *Lindholm et al., 2016*). However, the molecular mechanisms employed by most meiotic drivers are unknown.

The recently characterized *wtf* gene family of *Schizosaccharomyces pombe* includes several meiotic drivers (*Bravo Núñez et al., 2018a*; *Eickbush et al., 2019*; *Hu et al., 2017*; *López Hernández and Zanders, 2018*; *Nuckolls et al., 2017*). The *wtf* coding sequences are small (~1 kb) and encode autonomous drivers that specifically kill meiotic products (spores) that do not inherit the *wtf*⁺ allele from *wtf*⁺/*wtf*⁻ heterozygotes. These drivers carry out targeted spore destruction using two proteins: a poison (Wtf^poison) to which all spores are exposed, and an antidote (Wtf^antidote) which rescues only the spores that inherit the *wtf*⁺ allele (*Figure 1A and B*). The two proteins of a given driver are encoded on largely overlapping coding sequences, but the antidote contains ~45 additional N-terminal amino acids (*Figure 1A*). The small size and autonomy of the *wtf* drivers make them promising candidates for use in gene drive systems. It is important, however, to first understand more about the molecular mechanisms of the Wtf proteins and whether they are likely to be functional in other species.

Here, we investigate the mechanisms of *wtf* drive using the *wtf4* allele as a model. We demonstrate that the Wtf4 proteins are functional outside of gametogenesis and in the budding yeast *Saccharomyces cerevisiae*, despite over 350 million years since the two yeasts shared a common ancestor (*Hoffman et al., 2015*). We also show that the two Wtf4 proteins assemble into distinct aggregated forms. Wtf4^poison forms toxic aggregates that are dispersed throughout the cytoplasm. The Wtf4^antidote forms aggregates that are recruited to the vacuole and vacuole-associated inclusions and are largely non-toxic. When the two Wtf4 proteins are expressed together, the Wtf4^antidote and

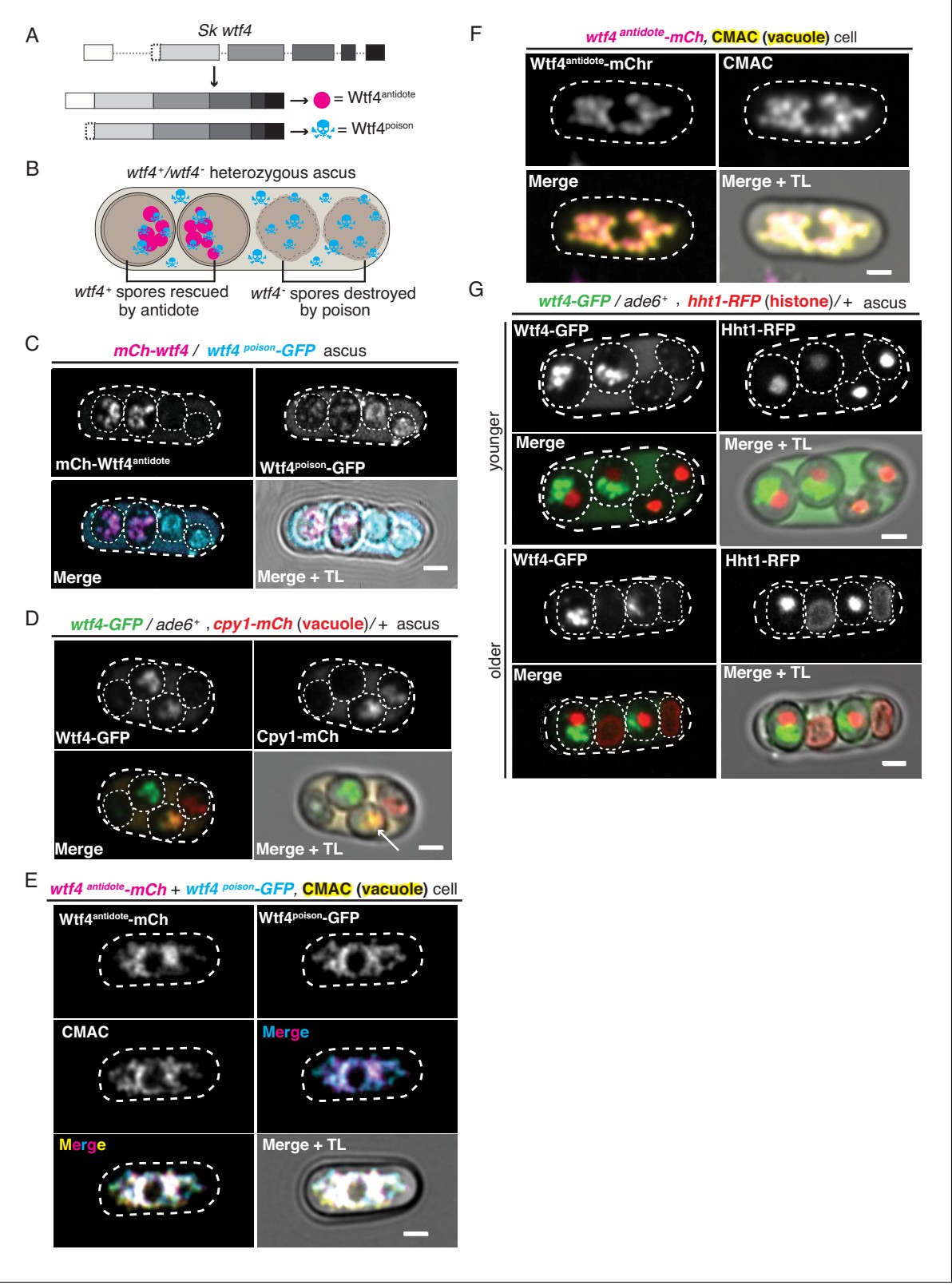

**Figure 1.** Wtf4$^{poison}$ and Wtf4$^{antidote}$ protein localization in both *S. pombe* meiosis and vegetative growth. (A) The *wtf4* gene utilizes alternate transcriptional start sites to encode for the Wtf4$^{antidote}$ and Wtf4$^{poison}$ proteins. (B) Model of a tetrad generated from a *wtf4$^{+}$/wtf4$^{-}$* diploid. *wtf4$^{+}$* spores are rescued by the spore-enriched antidote (magenta circles), while the poison (cyan skulls) spreads throughout the ascus. (C) An ascus generated by an *mCherry-wtf4/wtf4$^{poison}$-GFP* diploid showing the localization of mCherry-Wtf4$^{antidote}$ (magenta in merged images) and Wtf4$^{poison}$-GFP (cyan in merged

*Figure 1 continued on next page*

*Figure 1 continued*

images) (*Nuckolls et al., 2017*). (D) An ascus generated from a *wtf4-GFP/ade6⁺, +/cpy1* mCherry diploid showing localization of Wtf4-GFP (green in merged images) and Cpy1-mCherry (red in merged images). The arrow highlights the spore that inherited both tagged alleles and thus contains both tagged proteins. (E) A vegetatively growing haploid cell expressing Wtf4$^{poison}$-GFP (cyan in merged images) and Wtf4$^{antidote}$-mCherry (magenta in merged images) using the β-estradiol-inducible system. CMAC is a vacuole lumen stain (yellow in merged images). Both Wtf4 proteins colocalize with the vacuole. Cells were imaged 4 hr after induction with 100 nM β-estradiol. (F) A vegetatively growing haploid cell expressing Wtf4$^{antidote}$-mCherry (magenta in the merged images) using the β-estradiol system and stained with the CMAC vacuole stain (yellow in the merged images) shows Wtf4$^{antidote}$-mCherry in the vacuole. Cells were induced in the same way as in (E). (G) Asci generated from a *wtf4-GFP/ade6⁺, hht1-RFP/+* diploid. Hht1-RFP (red in merged images) is a histone marker. The nuclei in the spores that do not inherit *wtf4-GFP* (e.g. lacking GFP signal) can exhibit nuclear condensation and fragmentation. All scale bars represent 2 μm. TL = transmitted light.

The online version of this article includes the following figure supplement(s) for figure 1:

**Figure supplement 1.** Wtf4$^{poison}$ and Wtf4$^{antidote}$ protein sequences and colocalization of the proteins in *S. pombe* asci.
**Figure supplement 2.** Wtf4$^{poison}$ and Wtf4$^{antidote}$ colocalize in the vacuole and endoplasmic reticulum (ER) in spores.
**Figure supplement 3.** Wtf4$^{poison}$ and Wtf4$^{antidote}$ colocalize and are functional in vegetative *S. pombe* cells.
**Figure supplement 4.** Spores destroyed by Wtf4$^{poison}$ exhibit nuclear condensation followed by fragmentation.

Wtf4$^{poison}$ co-assemble and are trafficked to the vacuole. This work adds to our understanding of how *wtf* meiotic drivers work. In addition, the conserved function of Wtf4$^{poison}$'s toxicity and the fact that the Wtf4$^{antidote}$ exploits conserved aggregate management processes suggests that *wtf* genes represent good candidates for gene drive systems.

## Results

### Wtf4 proteins localize to the vacuole and endoplasmic reticulum within *S. pombe* spores

The *wtf4* meiotic driver used in this work is from *S. kambucha*, an isolate that is almost identical (99.5% DNA sequence identity) to the commonly studied lab isolate of *S. pombe* (*Rhind et al., 2011*; *Singh and Klar, 2002*). Our previous work demonstrated that the Wtf4$^{antidote}$ localizes to a region within the spores that inherit the *wtf4* gene. The Wtf4$^{poison}$ protein, however, is found in all four spores and throughout the sac (ascus) that holds them (*Nuckolls et al., 2017*). Here, we explored the localization of these proteins in greater depth to gain insight into their mechanisms.

We used fluorescently tagged alleles of *wtf4* to visualize the proteins. The two Wtf4 proteins have different translational start sites and thus different N-termini (*Figure 1A*, *Figure 1—figure supplement 1A*). We took advantage of this feature to visualize the proteins separately. For the Wtf4$^{antidote}$, we used an allele with an mCherry tag immediately upstream of the first start codon. This *mCherry-wtf4* allele tags only the Wtf4$^{antidote}$ (mCherry-Wtf4$^{antidote}$) but still encodes an untagged Wtf4$^{poison}$. We previously demonstrated that this allele is fully functional (*Nuckolls et al., 2017*). To visualize Wtf4$^{poison}$, we used the *wtf4$^{poison}$-GFP* allele. This separation-of-function allele encodes only a C-terminally tagged poison but no Wtf4$^{antidote}$ protein. We previously demonstrated that this tagged allele is functional but has a slightly weaker phenotype than an untagged *wtf4$^{poison}$* separation-of-function allele (*Nuckolls et al., 2017*).

We integrated the tagged alleles at the *ade6* locus in separate haploid *S. pombe* strains. We then crossed those two haploid strains to create heterozygous *mCherry-wtf4/wtf4$^{poison}$-GFP* diploids and induced these diploids to undergo meiosis. We imaged the asci using both standard and time-lapse fluorescence microscopy (*Figure 1C*, *Figure 1—figure supplement 1B*). We confirmed our previous observations that the mCherry-Wtf4$^{antidote}$ was enriched in two spores, whereas Wtf4$^{poison}$-GFP was found throughout the ascus and often formed puncta of various sizes. In the spores that did not inherit the antidote, Wtf4$^{poison}$-GFP also appeared dispersed throughout the spores. In the spores that inherited and thus expressed *mCherry-wtf4*, however, the localization of Wtf4$^{poison}$-GFP was more restricted. Specifically, we observed that the Wtf4$^{poison}$-GFP largely colocalized with mCherry-Wtf4$^{antidote}$ in a limited region of the spore (*Figure 1C*, *Figure 1—figure supplement 1B*). In time-lapse microscopy, it was evident that the two Wtf4 proteins consistently colocalized in a defined region of the spore, even as this region changed shape over time. This co-diffusion suggests the two proteins are either physically interacting or are present in the same compartment (*Figure 1—figure supplement 1B*). It also appeared that the level of Wtf4$^{poison}$-GFP protein is reduced in

spores containing the antidote. We did not distinguish if this was due to technical reasons (i.e. quenching of the GFP molecules) or biological reasons such as degradation of Wtf4[poison]-GFP in spores with mCherry-Wtf4[antidote] and/or due to a higher expression of Wtf4[poison]-GFP in the spores that inherit it (non-antidote spores) (*Figure 1C*). We completed Pearson correlation analysis (*Adler and Parmryd, 2010*) of mCherry-Wtf4[antidote] and Wtf4[poison]-GFP in the spores (where a result of >0 is positive correlation; 0, no correlation; <0 anti correlation) and obtained a coefficient of 0.61, indicating strong colocalization between the two Wtf4 proteins (*Figure 1—figure supplement 1C*).

The limited distribution of the Wtf4 poison and antidote proteins within *wtf4+* spores suggested they may be confined to a specific cellular compartment. To test this idea, we looked for colocalization of Wtf4 proteins with the vacuole, endoplasmic reticulum (ER) and nucleus (see below). For these experiments, we used the fully functional *wtf4-GFP* allele, which tags both the poison and antidote proteins (*Nuckolls et al., 2017*).

To assay the localization of the Wtf4 proteins relative to the vacuole, we imaged asci produced by diploids that were heterozygous for both *wtf4-GFP* and *cpy1-mCherry*. Cpy1-mCherry localizes to the lumen of the vacuole in vegetative cells (*Sun et al., 2013*) but has not, to our knowledge, been imaged in spores. We could observe mCherry in two of the four spores – presumably the two that inherited the *cpy1-mCherry* allele (*Figure 1D*). This 2:2 spore localization pattern has been previously observed in budding yeast for vacuolar proteins and several other organelles (*Neiman, 2011*; *Roeder and Shaw, 1996*; *Suda et al., 2007*). We found that the Wtf4-GFP and Cpy1-mCherry proteins colocalized within the spores that inherited both tagged alleles, suggesting the Wtf4 proteins are found within the vacuole (Pearson coefficient of 0.89, *Figure 1D*, *Figure 1—figure supplement 2A–B*). Interestingly, we also saw colocalization of Wtf4-GFP proteins with an ER marker, *pbip1-*mCherry-AHDL (*Zhang et al., 2012*; *Figure 1—figure supplement 2C–D*). We speculate this colocalization with the ER is due to nitrogen starvation which is required to induce meiosis and promotes organelle autophagy in *S. pombe* (*Kohda et al., 2007*; *Zhao et al., 2016*).

## Wtf4[antidote] localizes to the vacuole when its expression is induced in vegetatively growing *S. pombe* cells

Because we could not distinguish the vacuole and ER within spores, we assayed the localization of the Wtf4 proteins in the absence of nitrogen starvation. To do this, we fluorescently tagged the coding sequence of *wtf4[poison]* (*wtf4[poison]-GFP)* and *wtf4[antidote]* (*wtf4[antidote]-mCherry)* separation-of-function alleles under the control of β-estradiol-inducible promoters (*Ohira et al., 2017*). We then integrated the *wtf4[poison]-GFP* allele at the *ura4* locus and the *wtf4[antidote]-mCherry* allele at the *lys4* locus of the same haploid strain. Next, we observed the localization of the Wtf proteins relative to the vacuole (visualized using the CellTracker Blue CMAC lumen stain) or the ER (using Sec63-YFP) following β-estradiol induction. Similar to our observations in spores, we saw that the Wtf4[poison]-GFP and Wtf4[antidote]-mCherry proteins largely colocalized, with a Pearson coefficient of 0.68 (*Figure 1—figure supplement 3D and E*). We also found that the Wtf4 proteins colocalized with the CMAC stain (*Figure 1E*), which suggests that the Wtf4 poison and antidote proteins are largely within the vacuole. However, there were Wtf4[poison]-GFP puncta that lined the periphery of the cell and a circle in the middle of the cell, reminiscent of ER localization. These puncta were devoid of Wtf4[antidote]-mCherry (*Figure 1—figure supplement 3D*).

We also attempted to assay the localization of the Wtf4 antidote and poison proteins individually to test if the localization of the Wtf4[poison] was altered in the presence of the Wtf4[antidote], as we observed in spores (*Figure 1C*). We found that the localization of the Wtf4[antidote]-mCherry to the vacuole was similar in the absence of the Wtf4[poison] (*Figure 1F*, *Figure 1—figure supplement 3B*), with a Pearson coefficient of 0.69 (*Figure 1—figure supplement 3C*). This is analogous to previous observations of the localization of the slightly different Wtf4[antidote] protein (82.2% amino acid identity) found in the *S. pombe* lab strain (*Matsuyama et al., 2006*). We failed, however, to generate cells carrying the *wtf4[poison]-GFP* allele without the *wtf4[antidote]-mCherry* allele by transformation, or by crossing the strain carrying both *wtf4[poison]-GFP* and *wtf4[antidote]-mCherry* to a wild-type strain (*Figure 1—figure supplement 3A*). This is likely due to leaky expression of the *wtf4[poison]-GFP* from the inducible promoter even without addition of β-estradiol. Overall, our results suggest that the Wtf4[poison] protein is toxic in vegetative cells, but the antidote is still capable of neutralizing the poison, as we could obtain cells carrying both the Wtf4 poison and antidote proteins.

## S. pombe spores destroyed by *wtf4* display nuclear condensation followed by nuclear fragmentation

In the process of trying to understand the localization patterns of Wtf4 proteins, we assayed the localization of the Wtf4 proteins relative to the nucleus. For this experiment, we imaged asci produced by *wtf4-GFP/ade6+* heterozygotes also carrying a tagged histone allele, *hht1-RFP* (*Tomita and Cooper, 2007*). Although we did not observe colocalization of Wtf4 proteins and the nucleus, we frequently (24/38 asci) observed that the nuclei in the *wtf4-* spores appeared more condensed (*Figure 1G*, younger ascus). Additionally, in 11 out of 38 asci, one or both of the nuclei in the *wtf4-* spores were disrupted and the nuclear contents were dispersed throughout the spores (*Figure 1G*, older ascus). To address the timing of these nuclear phenotypes, we imaged diploids undergoing gametogenesis using time-lapse microscopy. We saw that all four nuclei tended to look similar shortly after the second meiotic division. As spores matured, however, we observed nuclear condensation sometimes followed by fragmentation in the spores that did not inherit *wtf4* (i.e. in spores lacking the enriched GFP expression and antidote function) (*Figure 1—figure supplement 4A and B*, see Materials and methods). This nuclear condensation and fragmentation are reminiscent of apoptotic cell death (*Carmona-Gutierrez et al., 2010*; *Kerr et al., 1972*).

## Wtf4 proteins function in the budding yeast, *Saccharomyces cerevisiae*

Our experiments in *S. pombe* suggest that the Wtf4 proteins can act when expressed outside of gametogenesis. However, our inability to induce expression of the Wtf4$^{poison}$ in the absence of the Wtf4$^{antidote}$ limited our ability to explore their mechanisms of action in this system. We, therefore, tested if the Wtf4 proteins functioned in the budding yeast *Saccharomyces cerevisiae*. To do this, we cloned the coding sequences of *wtf4$^{poison}$-GFP* and *wtf4$^{antidote}$-mCherry* under the control of β-estradiol inducible promoters on separate plasmids (*Ottoz et al., 2014*). We then introduced these plasmids into *S. cerevisiae* individually and together. We found that cells carrying the *wtf4$^{poison}$-GFP* plasmid were largely inviable when *wtf4$^{poison}$-GFP* expression was induced, indicating the poison is also toxic to *S. cerevisiae* (*Figure 2A*). However, cells expressing Wtf4$^{antidote}$-mCherry had only a slight growth defect relative to control cells carrying empty plasmids (*Figure 2A*). Importantly, expression of the Wtf4$^{antidote}$-mCherry plasmid largely ameliorated the toxicity of Wtf4$^{poison}$-GFP (*Figure 2A*). Given that *S. pombe* and *S. cerevisiae* diverged >350 million years ago (*Hoffman et al., 2015*), our results suggest that the target(s) of Wtf4$^{poison}$ toxicity are conserved and the Wtf4$^{antidote}$ does not require cofactors that are specific to *S. pombe* or gametogenesis to neutralize Wtf4$^{poison}$'s toxicity.

## Wtf4 poison and antidote proteins assemble into aggregates individually and together in budding yeast

We assayed the localization of the Wtf4 proteins in *S. cerevisiae* using the inducible *wtf4$^{poison}$-GFP* and *wtf4$^{antidote}$-mCherry* alleles described above. Similar to our observations in *S. pombe* gametogenesis, we saw that Wtf4$^{poison}$-GFP localized as puncta of varying sizes throughout the cytoplasm (*Figure 2B*). We also observed some of the Wtf4$^{poison}$-GFP protein localized to the ER (*Figure 2—figure supplement 1A*; *Friederichs et al., 2011*). Analogous to our observation in *S. pombe* spores, we saw nuclear condensation in cells expressing Wtf4$^{poison}$-GFP relative to wild-type cells (*Figure 2—figure supplement 1B–1D*).

Wtf4$^{antidote}$-mCherry, on the other hand, generally localized to one or two large amorphous regions adjacent to the vacuole (*Figure 2C*). When co-expressed, Wtf4$^{poison}$-GFP and Wtf4$^{antidote}$-mCherry co-localized to this region next to the vacuole (*Figure 2D*). In some cells, a faint circle of Wtf4$^{poison}$-GFP could also be observed (likely ER localization); however, the majority colocalized with the antidote in the vacuole-associated region (*Figure 2—figure supplement 1E*). This localization was similar but not identical to our observations in *S. pombe* cells, where the Wtf4 proteins localize within, rather than adjacent to, the vacuole. To ensure the difference in localization (sequestration to a single puncta) and cell viability of the Wtf4$^{poison}$-GFP protein observed in the cells co-expressing Wtf4$^{antidote}$-mCherry was not due to the mCherry tag, we also confirmed these results with an untagged Wtf4$^{antidote}$ (*Figure 2—figure supplement 2*).

Given that we failed to detect the Wtf proteins enter the vacuole, as we saw in *S. pombe*, we were interested in testing if the Wtf proteins were entering the vacuole but being rapidly degraded

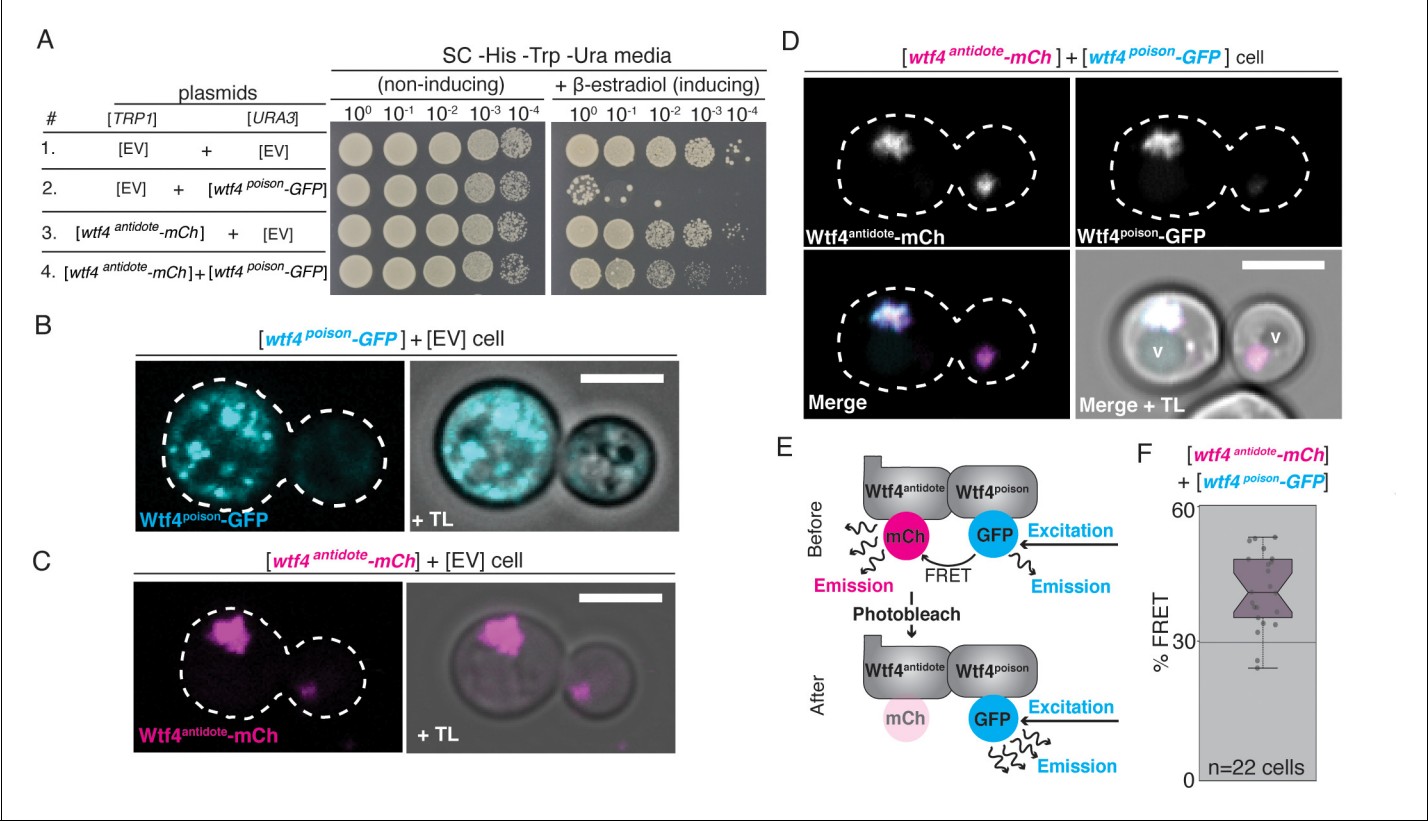

**Figure 2.** Wtf4<sup>poison</sup> and Wtf4<sup>antidote</sup> proteins physically interact and are functional in vegetative *S. cerevisiae* cells. (**A**) Spot assay of serial dilutions on non-inducing (SC -His -Trp -Ura) and inducing (SC -His -Trp -Ura + 500 nM β-estradiol) media. Each strain contains [*TRP1*] and [*URA3*] ARS CEN plasmids that are either empty (EV) or carry the indicated β-estradiol-inducible *wtf4* alleles. (**B**) A cell carrying an empty [*TRP1*] vector and a [*URA3*] vector with a β-estradiol inducible *wtf4<sup>poison</sup>-GFP* allele (cyan). (**C**) A haploid cell carrying an empty [*URA3*] vector and a [*TRP1*] plasmid with a β-estradiol inducible *wtf4<sup>antidote</sup>-mCherry* allele (magenta). (**D**) A haploid cell carrying a [*URA3*] plasmid with a β-estradiol inducible *wtf4<sup>poison</sup>-GFP* allele (cyan in merged images) and a [*TRP1*] plasmid with a β-estradiol inducible *wtf4<sup>antidote</sup>-mCherry* allele (magenta in merged images). The vacuole is marked with 'v'. (**E**) Cartoon of acceptor photobleaching Fluorescence Resonance Energy Transfer (FRET). If the two proteins interact, Wtf4<sup>poison</sup>-GFP (the donor) transfers energy to Wtf4<sup>antidote</sup>-mCherry (the acceptor). After photobleaching of the acceptor, the donor emission will increase. (**F**) Quantification of FRET values measured in cells carrying β-estradiol inducible Wtf4<sup>antidote</sup>-mCherry and β-estradiol inducible Wtf4<sup>poison</sup>-GFP. The cells showed an average of 40% FRET. These data are also presented in *Figure 5C*. In all experiments, the cells were imaged approximately 4 hr after induction in 500 nM β-estradiol. All scale bars represent 4 μm. TL = transmitted light.

The online version of this article includes the following figure supplement(s) for figure 2:

**Figure supplement 1.** Similarities in Wtf4 protein localization and Wtf4<sup>poison</sup>-induced cell death between *S. pombe* and *S. cerevisiae*.

**Figure supplement 2.** Untagged Wtf4<sup>antidote</sup> causes altered localization of Wtf4<sup>poison</sup>-GFP and neutralizes Wtf4<sup>poison</sup>-GFP toxicity in *S. cerevisiae*.

**Figure supplement 3.** Potential Wtf4<sup>antidote</sup>-mCherry and Wtf4<sup>poison</sup>-GFP degradation in *S. cerevisiae* cells.

**Figure supplement 4.** Wtf4<sup>poison</sup> and Wtf4<sup>antidote</sup> proteins assemble into aggregates at all detectable concentrations.

**Figure supplement 5.** Overexpression of various chaperones does not neutralize Wtf4<sup>poison</sup>-GFP toxicity.

in *S. cerevisiae*. To test this, we prepared protein samples from cells expressing Wtf4<sup>antidote</sup>-mCherry and Wtf4<sup>poison</sup>-GFP and performed western blots using α-mCherry and α-GFP antibodies. We analyzed protein from both the pellet and supernatant of our protein preparations because the predicted proteins are likely to be hydrophobic with six predicted transmembrane domains (*Figure 1—figure supplement 1A*) and their solubility is unknown (TMHMM model, *Krogh et al., 2001*). Using α-mCherry antibodies, we observed a prominent band at ~65 kDa that showed some smearing consistent with protein degradation. This band was not observed in samples prepared from cells not expressing Wtf proteins, suggesting it is likely full-length Wtf4<sup>antidote</sup>-mCherry protein (expected size 65 kDa). The band was more prominent in the pellet than the supernatant consistent with low solubility (*Figure 2—figure supplement 3*).

Using α-GFP antibodies, we observed multiple bands ranging from ~15–85 kDa, although the expected size of Wtf4$^{poison}$-GFP is 60 kDa. The bands were not observed in control samples prepared from cells not expressing Wtf proteins, suggesting the signal is specific to Wtf4$^{poison}$-GFP. The large apparent size of some of the Wtf4$^{poison}$-GFP bands suggest the protein may have post-translational modifications. In addition, the small size of some of the bands and the overall smeary appearance of the blot is consistent with degradation of the protein. Like the Wtf4$^{antidote}$-mCherry, a considerable amount of the Wtf4$^{poison}$-GFP was found in the pellet, suggesting both proteins have low solubility (*Figure 2—figure supplement 3*).

Given that the Wtf4$^{poison}$ and Wtf4$^{antidote}$ proteins colocalize and are both found in the pellet fraction of protein preparations, we tested if the proteins physically interact by using acceptor photobleaching Fluorescence Resonance Energy Transfer (FRET, *Sekar and Periasamy, 2003*) in cells expressing both Wtf4$^{poison}$-GFP and Wtf4$^{antidote}$-mCherry proteins. This process involves bleaching the fluorescence of a tagged protein (the acceptor) and looking for a corresponding increase in fluorescence of another tagged protein (the donor). If an increase in fluorescence of the donor is observed, the proteins are said to be physically interacting, as they are in close enough proximity (less than 10 nanometers) to transfer energy to each other (*Sekar and Periasamy, 2003*). When we bleached Wtf4$^{antidote}$-mCherry, we saw a corresponding increase in Wtf4$^{poison}$-GFP emission, supporting the idea that the two proteins physically interact (*Figure 2E and F*, *Figure 2—figure supplement 1F*).

The Wtf4 proteins localize as puncta of varying sizes, so we hypothesized that the proteins assemble into aggregates. To explore the nature of the Wtf4 protein assemblies, we utilized the recently developed Distributed Amphifluoric FRET (DAmFRET) assay (*Khan et al., 2018*). This approach looks for FRET between red and green versions of the same fluorophore in a partially photoconverted population of mEos3.1-tagged proteins as a measure of the protein's tendency to self-assemble (*Figure 2—figure supplement 4A*). We generated *wtf4$^{antidote}$-mEos3.1* and *wtf4$^{poison}$-mEos3.1* alleles, both under β-estradiol inducible promoters on *ARS CEN* plasmids. Both tagged constructs encoded functional proteins in *S. cerevisiae,* but the mEos3.1-tagged *wtf4$^{poison}$* allele was not as toxic as the GFP-tagged allele (*Figure 2—figure supplement 4B*). We then carried out DAmFRET analyses on cells producing either Wtf4$^{antidote}$-mEos3.1, Wtf4$^{poison}$-mEos3.1, or on cells producing both proteins simultaneously. We observed high ratiometric FRET signal (AmFRET) between Wtf4$^{antidote}$-mEos3.1 proteins and between Wtf4$^{poison}$-mEos3.1 proteins. In fact, all cells expressing Wtf4$^{poison}$-mEos3.1 and/or Wtf4$^{antidote}$-mEos3.1 proteins exhibited FRET as compared to mEos3.1 negative control, regardless of the expression level of the proteins (*Figure 2—figure supplement 4C*). The level of AmFRET did not change when both proteins were expressed simultaneously. Collectively, these experiments show that the Wtf4 proteins self-assemble, and that the proteins do not inhibit each other's assembly.

Interestingly, we did not detect any cells producing purely monomeric Wtf4$^{poison}$ in our DAmFRET analyses, indicating that self-assembly does not require a rate-limiting nucleation step that is characteristic of prion-forming proteins (*Khan et al., 2018*). This, combined with the irregular shape of the GFP puncta in our images, suggests that the toxic species of Wtf4$^{poison}$ is a poorly-ordered assembly of the protein. To explore this idea, we tested if overexpression of chaperones that promote protein homeostasis could neutralize Wtf4$^{poison}$ aggregates. First, we independently transformed multicopy 2-μm plasmids carrying various galactose-driven chaperones (*SIS1, YDJ1, HSP42, JJJ2, HSC82, HSP82, HSP104*) into a strain carrying the *wtf4$^{poison}$-mEos3.1* allele (*Figure 2—figure supplement 5*; *Li et al., 2016b*). Overexpression of any of these chaperones failed to ameliorate the toxicity of Wtf4$^{poison}$-mEos3.1, even when we reduced the levels of induction of the Wtf4$^{poison}$ (*Figure 2—figure supplement 5*). This suggests that neutralization of Wtf4$^{poison}$ may require the induction of multiple chaperones or stress response pathways at one time.

## Homotypic interactions promote co-assembly of Wtf4 proteins and neutralization of the Wtf4$^{poison}$

The Wtf4$^{poison}$ and Wtf4$^{antidote}$ proteins share the same 293 C-terminal amino acids (*Figure 1A*, *Figure 1—figure supplement 1A*). All the known active Wtf$^{antidote}$ proteins are highly similar to the Wtf$^{poison}$ they neutralize (*Bravo Núñez et al., 2020a*; *Bravo Núñez et al., 2020b*; *Hu et al., 2017*). In addition, mutations that disrupt the similarity between a given Wtf$^{antidote}$ and Wtf$^{poison}$ can eliminate the ability of the Wtf$^{antidote}$ to neutralize the Wtf$^{poison}$ (*Bravo Núñez et al., 2018a*). Here, we

tested the mechanism underlying that requirement using Wtf4 proteins. Given that each Wtf4 protein self-assembles (*Figure 2—figure supplement 4*), we hypothesized that homotypic interactions between Wtf4[poison] and Wtf4[antidote] mediate their co-assembly and neutralization of the poison. To test this idea, we mutated sequences at the C-termini of the inducible *wtf4[poison]-GFP* and *wtf4[antidote]-mCherry* alleles in the *S. cerevisiae* plasmids described above. Specifically, we targeted our mutagenesis to a seven amino acid sequence (IGNAFRG) that is found in varying copy numbers in many members of the *wtf* gene family (*Eickbush et al., 2019*). We previously showed that a mismatched number of these repeats between Wtf poison and antidote proteins is enough to disrupt their specificity (*Bravo Núñez et al., 2018a*). The wild-type *S. kambucha wtf4* allele contains ~1.5 repeat units (*Figure 3A*). To make the mutants, we inserted 18 additional codons into the repeat region of *wtf4* to make a total of four repeats. We denote these repeat insertion mutants with an * (*Figure 3A*).

As expected, the Wtf4[poison]*-GFP protein is functional (i.e. toxic) in *S. cerevisiae* and localizes similarly to the tagged wild-type Wtf4[poison]-GFP (*Figure 3B*, *Figure 3—figure supplement 1A*). Wtf4[poison]*-GFP is neutralized by the matching Wtf4[antidote]*-mCherry protein, and the two mutant proteins colocalized as vacuole-associated assemblies, just like the tagged wild-type proteins in *S. cerevisiae* (*Figure 3B and C*). The Wtf4[antidote]*-mCherry protein on its own also resembled the wild-type Wtf4[antidote]-mCherry protein localization (*Figure 3—figure supplement 1B*). The Wtf4[antidote]*-mCherry could not, however, suppress the toxicity of the wild-type Wtf4[poison]-GFP (*Figure 3B*). Similarly, the wild-type Wtf4[antidote]-mCherry could not neutralize the Wtf4[poison]*-GFP's toxicity (*Figure 3B*). The poison and antidote proteins did not colocalize in cells with incompatible poison and antidote proteins, and instead the poison proteins formed distributed aggregates, similar to cells expressing no antidote (*Figure 3D and E*).

## Electron microscopy reveals an association between Wtf4 aggregates and vesicles in *S. cerevisiae*

We next used transmission electron microscopy (TEM) to analyze the environment of Wtf proteins within the vacuole-associated aggregates. Similar to our observations made using fluorescence microscopy, we found using immuno-gold labeling that Wtf4[poison]-GFP largely clustered near the vacuole in cells also producing untagged Wtf4[antidote] (*Figure 4A*). These images also revealed that the Wtf4 protein aggregates appeared within a cluster of lightly staining organelles resembling lipid droplets (*Figure 4A*, *Figure 4—figure supplement 1A*). Very few immunogold particles were found in the cells carrying only empty vectors, suggesting minimal background and high specificity of the GFP antibody used for the immunolabeling (*Figure 4—figure supplement 1B*).

To look at these Wtf aggregate-associated organelles at higher resolution, we used TEM with a sample preparation method that better maintains cellular morphology (see Materials and methods). We found that the organelles were in fact a mix of lipid droplets and large vesicles with bilayer membranes (*Figure 4B* arrows, *Figure 4—figure supplement 2A–C*). We quantified the number of lipid droplets and large vesicles in cells carrying empty vectors and in cells carrying both β-estradiol inducible Wtf4[antidote] and β-estradiol inducible Wtf4[poison]-GFP. We found that cells expressing Wtf4 proteins had significantly more lipid droplets and large vesicles (*Figure 4D*, *Figure 4—figure supplement 2F*). These results indicate that the large aggregates that form in cells expressing Wtf4[antidote] are embedded in a cluster of large vesicles and lipid droplets. This phenotype is reminiscent of another aggregation prone protein, α-synuclein, a protein associated with Parkinson's disease in humans, that when expressed in yeast forms cytoplasmic accumulations in association with clusters of vesicles (*Soper et al., 2008*). The α-synuclein vesicles, however, appear smaller and more numerous than the Wtf4-associated vesicles. To test if the increase in vesicles and lipid droplets was a common feature of aggregation prone proteins, we expressed (using the β-estradiol system) a different vacuole-associate prion aggregate, Rnq1-mCardinal (*Figure 4—figure supplement 2E*). We did not observe elevated vesicles or lipid droplets, suggesting that the increase in vesicles is due to the Wtf4 aggregates, not a consequence of the over-expression system or a general feature of aggregation prone proteins.

We also imaged cells expressing only Wtf4[antidote] or Wtf4[poison]. The morphology of cells expressing only the Wtf4[antidote] was indistinguishable from cells expressing both Wtf proteins (*Figure 4—figure supplement 2D*). It is difficult to interpret the observations from the (dying) cells expressing only Wtf4[poison] because the majority of the cells did not maintain cellular integrity during sample

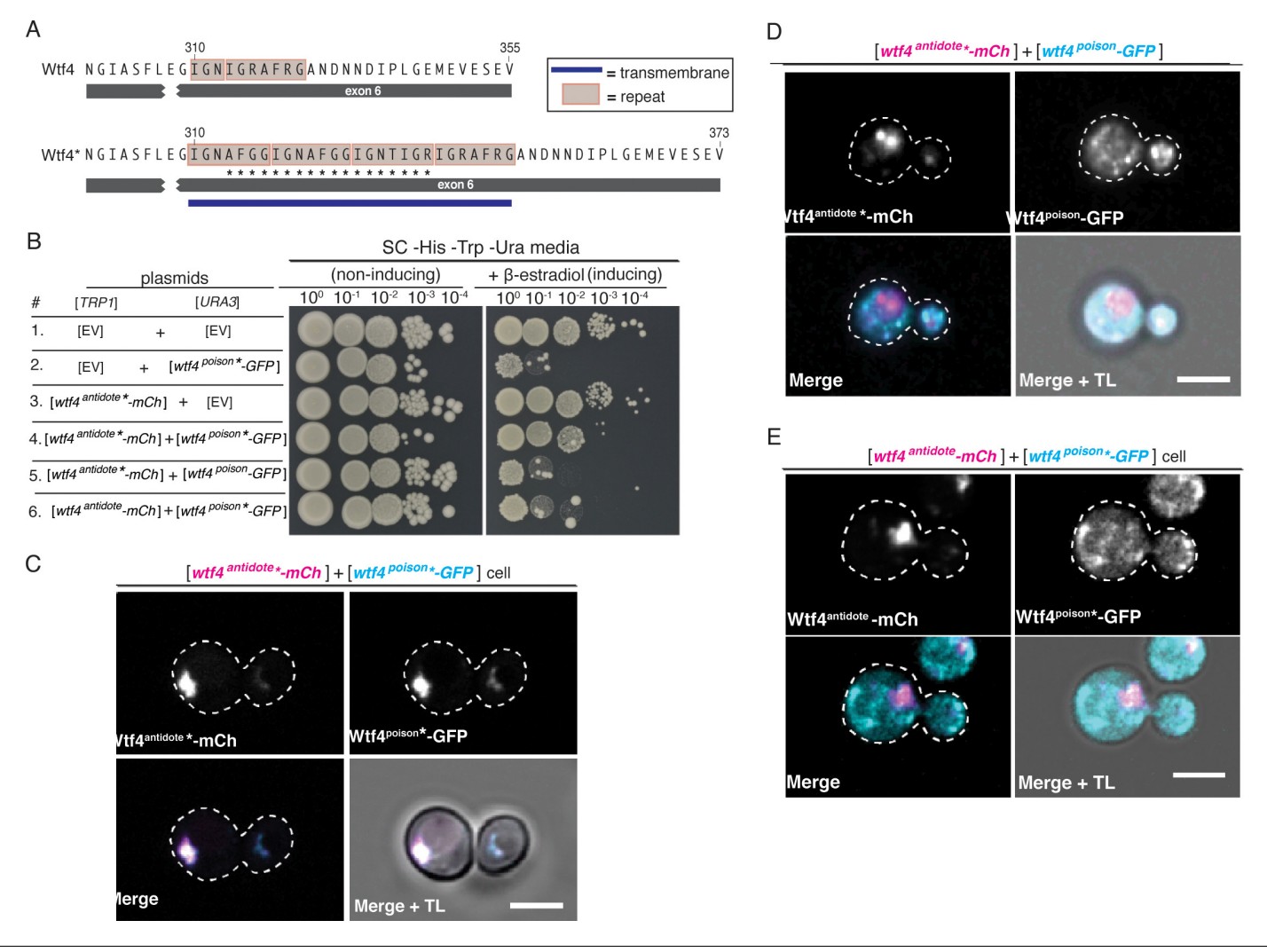

**Figure 3.** Homotypic interactions facilitate Wtf4poison-Wtf4antidote co-assembly and Wtf4antidote function. (A) DNA sequence encoding 18 amino acids (marked with an *) were added to the repeat sequences within exon 6 of the *wtf4* allele to create *wtf4** alleles. (B) Spot assay of serial dilutions on non-inducing (SC -His -Trp -Ura) and inducing (SC -His -Trp -Ura + 500 nM β-estradiol) media. Each strain contains [*TRP1*] and [*URA3*] ARS CEN plasmids that are either empty (EV) or carry the indicated β-estradiol inducible *wtf4* alleles. (C) Representative image of a haploid cell carrying a [*TRP1*] ARS CEN plasmid with a β-estradiol inducible *wtf4antidote*-mCherry* allele (magenta in merged images) and a [*URA3*] ARS CEN plasmid with a β-estradiol inducible *wtf4poison*-GFP* (cyan in merged images). (D) Representative image of a haploid cell carrying a [*TRP1*] ARS CEN plasmid with a β-estradiol inducible *wtf4antidote*-mCherry* allele (magenta in merged images) and a [*URA3*] ARS CEN plasmid with a β-estradiol inducible *wtf4poison-GFP* allele (cyan in merged images). (E) Representative image of a haploid cell carrying a [*TRP1*] ARS CEN plasmid with a β-estradiol inducible *wtf4antidote-mCherry* allele (magenta in merged images) and a [*URA3*] ARS CEN plasmid with a β-estradiol inducible *wtf4poison*-GFP* allele (cyan in merged images). In all experiments, the cells were imaged ~4 hr after induction in 500 nM β-estradiol. All scale bars represent 4 μm. TL = transmitted light.

The online version of this article includes the following figure supplement(s) for figure 3:

**Figure supplement 1.** Wtf4poison* and Wtf4antidote* have the same localization as the wild-type Wtf4 proteins in *S. cerevisiae*.

preparation (*Figure 4—figure supplement 3A*). In the few cells we could image, we observed diverse morphologies. We generally did not observe clustering of large vesicles and lipid droplets, as we saw in cells expressing Wtf4antidote. Instead, organelle integrity often looked disrupted and many cells expressing Wtf4poison appeared to have undergone extensive autophagy (*Figure 4—figure supplement 3B–D*).

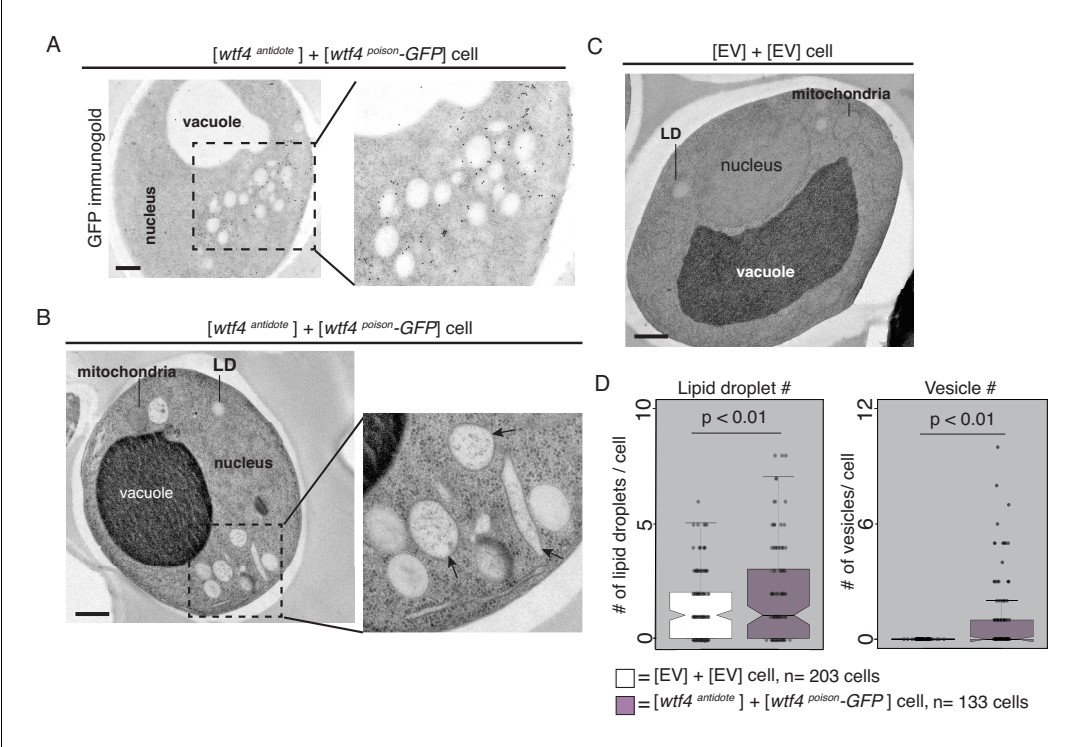

**Figure 4.** Cells expressing Wtf4[poison] and Wtf4[antidote] or Wtf4[antidote] have large vesicles that cluster within the Wtf4 aggregate. (**A**) Representative tomograph of immuno-gold Transmission Electron Microscopy (TEM) of a haploid cell carrying a [*TRP1*] *ARS CEN* plasmid with a β-estradiol inducible *wtf4[antidote]* allele and a [*URA3*] *ARS CEN* plasmid with a β-estradiol inducible *wtf4[poison]*-*GFP* allele. A monoclonal antibody against GFP was used. Immunogold particles (black dots) are enriched in a cluster near light staining organelles. (**B**) Representative TEM tomograph of cells of the same genotype as in (**A**). Arrows point to vesicle structures. (**C**) Representative TEM tomograph of a cell carrying an empty [*TRP1*] vector and an empty [*URA3*] vector. (**D**) Quantification of the number of lipid droplets (left) or vesicles (right) per cell of two samples: 1. Cells carrying empty [*TRP1*] and [*URA3*] vectors (EV, white, n = 203 cells) and 2. Cells carrying a [*TRP1*] *ARS CEN* plasmid with a β-estradiol inducible *wtf4[antidote]* allele and a [*URA3*] *ARS CEN* plasmid with a β-estradiol inducible *wtf4[poison]*-*GFP* allele (purple, n = 133 cells), (p<0.01, t-test). All samples were processed ~4 hr after induction in 500 nM β-estradiol. All scale bars represent 0.5 μm. Lipid droplet = LD.

The online version of this article includes the following figure supplement(s) for figure 4:

**Figure supplement 1.** Cells expressing Wtf4[poison] and Wtf4[antidote] or Wtf4[antidote] have large vesicles that cluster within the Wtf4 aggregate.

**Figure supplement 2.** Cells expressing Wtf4[poison]+Wtf4[antidote] or Wtf4[antidote] have large vesicles that cluster within the Wtf4 aggregate.

**Figure supplement 3.** Cells producing only Wtf4[poison] have variable phenotypes.

## Wtf4 poison–antidote protein aggregates often localize to the IPOD and PAS in budding yeast

Our results were reminiscent of other studies in which toxic aggregated proteins were neutralized via sequestration at cellular inclusions (*Kaganovich et al., 2008*; *Liu et al., 2010*; *Taylor et al., 2003*; *Chen et al., 2011*; *Hill et al., 2017*; *Tyedmers et al., 2010*; *Kryndushkin et al., 2012*; *Bagola and Sommer, 2008*; *Arrasate et al., 2004*). In *S. cerevisiae*, stable, misfolded proteins are generally sequestered to the Insoluble PrOtein Deposit (IPOD), a compartment located near the vacuole and pre-autophagosomal site (PAS) (*Kaganovich et al., 2008*; *Tyedmers et al., 2010*; *Suzuki and Ohsumi, 2010*; *Rothe et al., 2018*). This compartmentalization of damaged/misfolded proteins mitigates their toxic effects and facilitates their disposal, some of which occurs via autophagy (*Marshall et al., 2016*).

Given that the Wtf4[antidote] and Wtf[poison]+Wtf4[antidote] aggregates localize adjacent to the vacuole, we hypothesized that they could be at the IPOD in *S. cerevisiae*. To test this idea, we looked for the localization of the Wtf4 proteins relative to Rnq1-mCardinal and GFP-Atg8. Rnq1 localizes to the IPOD and Atg8 is a component of the pre-autophagosomal structure that is adjacent to the IPOD (*Kaganovich et al., 2008*; *Tyedmers et al., 2010*; *Rothe et al., 2018*). Consistent with our

hypothesis, we found that Wtf4antidote-mCherry either colocalized or was adjacent to Rnq1-mCardinal (*Figure 5A*, *Figure 5—figure supplement 1A–1C*). Wtf4poison-GFP did not colocalize with Rnq1-mCardinal on its own, supporting the idea that Wtf4antidote recruits the poison to the IPOD (*Figure 5—figure supplement 1D*).

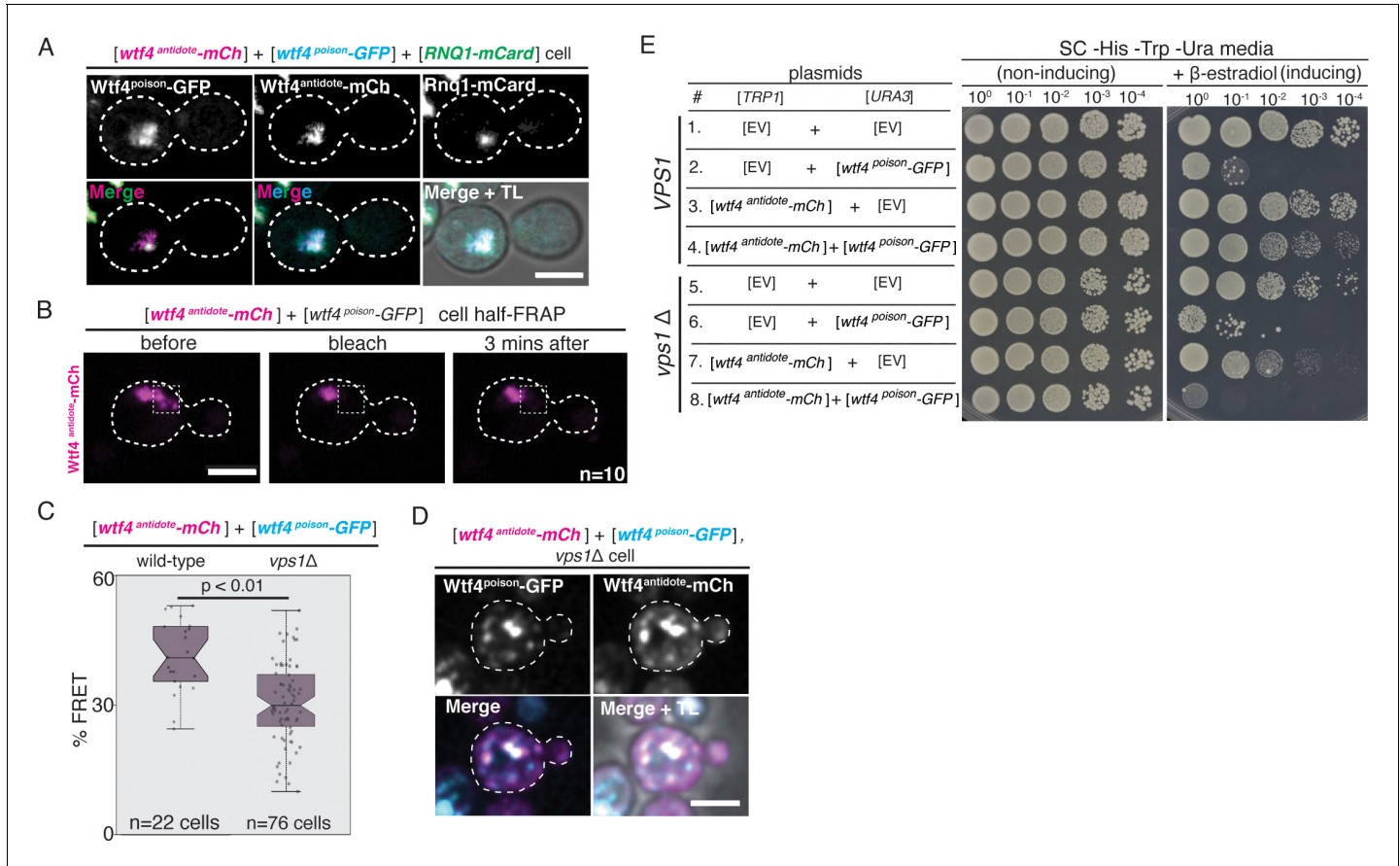

**Figure 5.** The Wtf4 proteins colocalize at the Insoluble Protein Deposit (IPOD). (**A**) Representative image of a vegetatively growing haploid cell carrying a [*TRP1*] vector with a β-estradiol inducible *wtf4antidote-mCherry* (magenta in merged images) allele, a [*URA3*] vector with a β-estradiol inducible *wtf4poison-GFP* (cyan) allele, and a [*LEU2*] vector with a β-estradiol inducible *RNQ1-mCardinal* allele, acting as an IPOD marker (green in merged images). (**B**) half-Fluorescence Recovery After Photobleaching (half-FRAP) of the Wtf4antidote-mCherry aggregate in cells carrying a [*TRP1*] vector with a β-estradiol inducible *wtf4antidote-mCherry* (magenta) allele and a [*URA3*] vector with a β-estradiol inducible *wtf4poison-GFP* allele. Cells were imaged for 3 min after bleaching and no recovery of mCherry fluorescence was observed. (**C**) Quantification of FRET values of Wtf4antidote-mCherry and Wtf4poison-GFP measured in wild-type (same data as *Figure 2F*) and *vps1Δ* cells carrying vectors with β-estradiol inducible *wtf4antidote-mCherry* allele and β-estradiol inducible *wtf4poison-GFP* allele (p>0.01, t-test). (**D**) Representative image of a vegetatively growing, haploid *vps1Δ* cell carrying a [*URA3*] vector with a β-estradiol inducible *wtf4poison-GFP* allele (cyan in merged images) and a [*TRP1*] vector with a β-estradiol inducible *wtf4antidote-mCherry* (magenta in merged images) allele. All fluorescence microscopy images acquired after ~4 hr in 500 nM β-estradiol media. All scale bars represent 4 µm. TL = transmitted light. (**E**) Spot assay of serial dilutions on non-inducing inducing (SC -His -Trp -Ura) and inducing (SC -His -Trp -Ura + 500 nM β-estradiol) media of both wild-type (top, samples 1–4) and *vps1Δ* (bottom, samples 5–8) cells. Each strain contains [*TRP1*] and [*URA3*] ARS CEN plasmids that are either empty (EV) or carry the indicated β-estradiol-inducible *wtf4* alleles.

The online version of this article includes the following source data and figure supplement(s) for figure 5:

**Figure supplement 1.** The Wtf4 proteins colocalize at the Insoluble Protein Deposit (IPOD).

**Figure supplement 2.** A screen of the *S. cerevisiae* deletion collection to identify genes necessary for survival upon Wtf4poison and Wtf4antidote expression.

**Figure supplement 2—source data 1.** Results of a genetic screen of the *S. cerevisiae* deletion collection to identify genes necessary for survival upon Wtf4 protein expression.

**Figure supplement 3.** Overexpression *of ATG8* causes vacuolar localization of Wtf4antidote-mCherry.

**Figure supplement 4.** Wtf4antidote-mCherry is stable in *S. pombe* spores.

**Figure supplement 5.** Vps1 is necessary for the recruitment of Wtf4 proteins to the Insoluble Protein Deposit (IPOD).

To visualize the localization of Wtf4[antidote] relative Atg8, we expressed Wtf4[antidote]-mCherry from a Gal-driven promoter on a plasmid. This is a different expression system than our other experiments described so far, but the protein is functional (*Figure 5—figure supplement 2A–B*). We found that Wtf4[antidote]-mCherry colocalized or was adjacent to GFP-Atg8 (expressed from a plasmid under its endogenous promoter) when both proteins localized outside the vacuole (*Figure 5—figure supplement 3D I*) *Guan et al., 2001*. This supports the idea that Wtf4[antidote] localizes at the IPOD. However, in the majority of cells expressing both Wtf4[antidote]-mCherry and GFP-Atg8, the proteins both localized inside the vacuole (*Figure 5—figure supplement 3D, F H*). This was different than cells expressing either protein alone, where Atg8-GFP localizes as a focus outside the vacuole (*Figure 5—figure supplement 3E*) and Wtf4[antidote]-mCherry localizes as a large aggregate outside the vacuole (*Figure 5—figure supplement 3G*). This suggests that the overexpression of *GFP-ATG8* enhances the degradation of Wtf4[antidote]-mCherry. Additionally, these results could suggest that the increased vacuolar localization of GFP-Atg8 in cells producing Wtf4[antidote]-mCherry is due to enhanced recruitment and degradation of autophagosomes. Indeed, we saw increased vesicles in cells expressing Wtf4[antidote] (*Figure 4*).

Proteins in the IPOD tend to be insoluble (*Bagola and Sommer, 2008*; *Kaganovich et al., 2008*). To test if the Wtf4[antidote] shared this property in *S. cerevisiae*, we used half punctum-Fluorescence Recovery After Photobleaching (half-FRAP) (*Khan et al., 2018*; *Zhang et al., 2015*). This analysis revealed that the Wtf4[antidote]-mCherry aggregate has very low internal mobility and is thus more solid-like than liquid-like (*Figure 5B*). We were curious if the Wtf4[antidote] behaved similarly in its native context. To test this, we performed the half-FRAP assay on the Wtf4[antidote]-mCherry in *S. pombe* spores and, consistent with our result in *S. cerevisiae,* found very low protein mobility (*Figure 5—figure supplement 4A B*).

## Genes involved in mitochondrial function, stress response pathways, and vesicle trafficking are necessary to neutralize Wtf4 protein toxicity in budding yeast

To better understand how toxic Wtf4 protein aggregates are neutralized, we screened for genes necessary for survival after induction of *wtf4[antidote]* and *wtf4[poison]*. Briefly, we screened the *S. cerevisiae MATa,* haploid deletion collection for mutants that failed to survive on galactose media when they carried plasmids containing galactose-inducible *wtf4[antidote]-mCherry* and *wtf4[poison]-GFP* genes (*Figure 5—figure supplement 2A B*). We found 106 mutants that could grow on galactose when carrying empty vector plasmids, but not when carrying both *wtf4* plasmids (*Figure 5—figure supplement 2—source data 1*). We also provide results of similar screens for comparison (*Figure 5—figure supplement 2—source data 1*; *Willingham et al., 2003 Enyenihi and Saunders, 2003*).

Amongst our hits, the only significantly enriched (FDR p<0.05) gene ontology groups were mitochondrial translation and organization (*Figure 5—figure supplement 2—source data 1*). We speculate this enrichment is due to two known roles of mitochondria in managing protein aggregates. The first is the Mitochondria As Guardian In Cytosol (MAGIC) mechanism in which mitochondria help degrade protein aggregates (*Ruan et al., 2017*). The second is that mitochondria mitigate the impact of toxic aggregates by promoting asymmetric aggregate segregation in mitosis (*Zhou et al., 2014*).

We also identified genes involved in cell wall integrity pathways (*POP2, MPT5, SLT2* and *BCK1*) as necessary for survival after induction of Wtf4[antidote] and Wtf4[poison] (*Jin et al., 2015*; *Li et al., 2016a*; *Stewart et al., 2007*). The cell wall integrity pathway is triggered by diverse stress stimuli (*Fuchs and Mylonakis, 2009*) and can promote stress-response gene expression and nuclear release of cyclin-C ((Ssn8), also a hit in our screen) (*García et al., 2009*). Release of cyclin-C into the cytoplasm promotes mitochondrial hyper-fission, stress response gene activation, and either apoptosis or repair of the stress-induced damage (*Jin et al., 2015*). Consistent with this, we observed separated, significantly smaller mitochondria in cells expressing the Wtf4 proteins (*Figure 4—figure supplement 2C and G*). Altogether, our screen hits suggest links between the cell wall integrity stress response pathways, mitochondrial fission, and Wtf4[antidote] function.

Several other screen hits were genes with known roles in maintaining protein homeostasis and/or aggregate management. For example, we found that several genes involved in vesicle transport, endocytosis, and trafficking to the vacuole (e.g. *ATG11, SNF7,* and multiple *VPS* genes) are also required for survival when the Wtf4 proteins are expressed (*Figure 5—figure supplement 2—*

*source data 1*). These hits suggest vacuolar trafficking pathways contribute to the neutralization of Wtf4 protein aggregates. This is consistent with our EM analyses showing that the Wtf4antidote inclusion site is enriched with vesicles. Previous work demonstrated these pathways are also important for trafficking other proteins to the IPOD and for neutralizing the toxicity of the aggregation prone TDP-43 (*Rothe et al., 2018*; *He et al., 2006*; *Liu et al., 2017*).

Given our results, which suggest that Wtf4 protein localization is an important factor in mitigating toxicity, we next imaged the localization of Wtf4poison-GFP and Wtf4antidote-mCherry in all of the screen hits. We found that the localization of the Wtf4poison-GFP and Wtf4antidote-mCherry proteins was disrupted in all 106 hits relative to wild type (where the proteins coalesce to the IPOD). In 81 mutants, the Wtf4poison-GFP and Wtf4antidote-mCherry proteins localized as dispersed aggregates throughout the cell. These mutants included deletions of *YNL170W*, a reported dubious open reading frame, and *PHD1*, a transcriptional activator (*Figure 5—figure supplement 2C D*). We noted there were often cells with dispersed Wtf4antidote-mCherry aggregates or cells with dispersed Wtf4poison-GFP aggregates, but rarely cells with both. We speculate this is due to toxicity of distributed aggregates and cells expressing both aggregates at the same time being destroyed quickly. Another common feature we observed throughout the screen hits was Wtf4antidote-mCherry signal in the vacuole. We also observed this vacuolar localization in the C-terminal mutants depicted in *Figure 3D and E*, so this appears to be a common feature of the Wtf4antidote-mCherry protein in cells being destroyed by Wtf4poison. Five mutants appeared to have wild-type looking Wtf4antidote (single inclusion outside the vacuole) but dispersed Wtf4poison, suggesting that the mutations may disrupt the interaction of the poison and antidote (*Figure 5—figure supplement 2E F*). Twenty hits showed very little Wtf4 signal and soluble cytoplasmic localization of both proteins. (*Figure 5—figure supplement 2G*).

Because the Wtf4antidote protein is quite similar to the Wtf4poison and also assembles into aggregates, we were curious if the Wtf4antidote alone was toxic in the absence of any of our screen hits. We therefore assayed the viability of the 106 deletion mutants when only Wtf4antidote was expressed. We saw that in approximately half (44/106) of the deletion stains Wtf4antidote expression reduced viability (*Figure 5—figure supplement 2—source data 1*). These results are consistent with the idea that active aggregate management pathways are often required for cells to mitigate the toxicity of even the Wtf4antidote protein in *S. cerevisiae*.

We also investigated one hit from our screen, *VPS1*, more thoroughly using our β-estradiol-inducible *wtf4* plasmids (described above). Vps1 is a dynamin-like GTPase that is necessary for trafficking of aggregates to the IPOD and/or other inclusion sites (*Kumar et al., 2016*; *Kumar et al., 2017*; *Hill et al., 2016*; *Marshall et al., 2016*). In the absence of *VPS1*, we found that the Wtf4antidote-mCherry and Wtf4poison-GFP proteins still physically interact (*Figure 5C*, *Figure 5—figure supplement 5A*). The Wtf4 protein aggregates did not, however, coalesce to form large inclusions (*Figure 5D*, *Figure 5—figure supplement 5A and B*) and Wtf4antidote-mCherry failed to neutralize the toxicity of Wtf4poison-GFP in *vps1Δ* cells (*Figure 5E*).

Together, these experiments indicate that the physical interaction between the Wtf4poison and Wtf4antidote proteins is insufficient to neutralize the toxicity of Wtf4poison protein aggregates. Trafficking the aggregates to a vacuole-associated inclusion is also required. Interestingly, we also observed enhanced toxicity of the Wtf4antidote-mCherry protein in the absence of Vps1 and many of our other screen hits (*Figure 5E*, *Figure 5—figure supplement 2—source data 1*). These results suggest that the antidote aggregates are more detrimental to cells when they are distributed in the cytoplasm. Importantly, however, even in the *vps1Δ* mutant, expression of Wtf4antidote-mCherry is less toxic to cells than Wtf4poison-GFP. This, and the fact that not all of the 106 hits caused Wtf4antidote-mCherry to become toxic, suggests there are fundamental differences in the poison and antidote aggregates beyond their propensity to be trafficked to a vacuole-associated inclusion.

## Discussion

### Wtf4 proteins exploit conserved aspects of cell physiology to cause selective cell death

Here, we explored how the Wtf4poison protein kills cells and how the Wtf4antidote protein neutralizes the toxicity of the Wtf4poison. We used a combination of genetics and cell biology to study these

proteins in three contexts: (1) their endogenous context of *S. pombe* gametogenesis, (2) vegetatively growing *S. pombe* cells, and (3) vegetatively growing *S. cerevisiae* cells. In all three contexts, expression of Wtf4$^{poison}$ alone kills cells and expression of the Wtf4$^{antidote}$ rescues the toxicity. The simplest interpretation of these observations is that Wtf4$^{poison}$ exploits or disrupts a conserved aspect of cellular physiology that is important during both vegetative growth and gametogenesis. Similarly, the Wtf4$^{antidote}$ neutralizes the Wtf4$^{poison}$ using conserved cofactors that can act in both vegetative growth and gametogenesis. This conservation suggests that *wtf*-derived gene drives could be a useful tool for genetically altering populations.

## Wtf4$^{poison}$ proteins assemble into toxic aggregates

In *S. pombe* gametogenesis and in vegetative *S. cerevisiae* cells, we observed the Wtf4$^{poison}$-GFP proteins assembled into small foci (aggregates) in the absence of Wtf4$^{antidote}$. The aggregates were largely dispersed throughout the cytoplasm, with some ER localization. The assembly of Wtf4 proteins is reminiscent of another meiotic drive element, Het-s, which employs prion-like amyloid polymerization to convert Het-S proteins to a lethal form (*Dalstra et al., 2003*; *Riek and Saupe, 2016*). We therefore evaluated whether Wtf4$^{poison}$ proteins exhibit prion activity in *S. cerevisiae* using DAmFRET (*Khan et al., 2018*). We found that Wtf4$^{poison}$-mEos proteins assembled with themselves even at very low expression levels (*Figure 2—figure supplement 4C*). In fact, we were unable to detect cells that lacked self-assemblies, revealing that the toxic form of the protein is not appreciably supersaturated, as would be required for Wtf4$^{antidote}$ to detoxify it through a simple prion-like mechanism where the antidote templates the deposition of poison monomers. Nevertheless, the sequence-dependent self-assembly of Wtf4 remains consistent with amyloid polymerization. However, given its intimate association with vesicles, extensive testing would be required to further evaluate the structural basis of Wtf4 activity.

The significance of the Wtf4$^{poison}$ aggregation is not clear. We speculate that the aggregation propensity is intimately tied to the toxicity of Wtf4$^{poison}$. We propose that distributed Wtf4 aggregates interact broadly with other proteins and disrupt their folding or localization. Compounding effects of these hypothesized interactions could disrupt protein homeostasis or cellular integrity, leading to cell death. This death may occur via a programmed cell death pathway, as in both *S. pombe* gametogenesis and in vegetative *S. cerevisiae,* cells succumbing to the Wtf4$^{poison}$ exhibit nuclear condensation (followed by nuclear fragmentation in *S. pombe*). The death may also be related to loss of cell wall integrity, as cell wall integrity pathways are necessary for cell survival upon expression of the Wtf4 proteins. Testing these ideas may be challenging, especially if understanding Wtf4$^{poison}$ toxicity proves to be as elusive as understanding the intensely studied neurotoxic aggregating proteins TDP-43 and α-Synuclein (*Johnson et al., 2009*; *Cookson and van der Brug, 2008*).

## Wtf4$^{antidote}$ promotes neutralization of Wtf4$^{poison}$ via recruitment to vacuole-associated sites

Like Wtf4$^{poison}$, the Wtf4$^{antidote}$ also assembles into aggregates in both *S. pombe* and *S. cerevisiae* cells. Unlike the Wtf4$^{poison}$, however, the Wtf4$^{antidote}$ aggregates have little effect on the viability of wild-type vegetative cells. This is surprising given the similarity of the two proteins (the Wtf4$^{poison}$ shares 292 of the Wtf4$^{antidote}$'s 337 amino acids). Our data suggest that the localization of the aggregates and/or the exposed aggregate surface area could underlie their differences in toxicity. The Wtf4$^{poison}$ aggregates (without Wtf4$^{antidote}$) remain largely dispersed in the cytoplasm, whereas the Wtf4$^{antidote}$ proteins are trafficked to a confined region near or within the vacuole.

In *S. pombe* cells, the Wtf4$^{antidote}$ aggregates enter the vacuole. In *S. cerevisiae* cells, the Wtf4$^{antidote}$ accumulates outside the vacuole in the IPOD, but could also be trafficked into the vacuole at some rate. We observed signs of protein degradation, but this could be due to vacuolar degradation or other degradation mechanisms (*Figure 2—figure supplement 3*). The different modes of cell division in *S. pombe* and *S. cerevisiae* may contribute to the differences between how the species handle Wtf4 aggregates. *S. cerevisiae* divides asymmetrically by budding and tends to retain aggregates, including Wtf4 proteins, in the mother cells (*Spokoini et al., 2012*). *S. pombe*, however, generally divides symmetrically making it difficult for new cells to be born free of aggregates found in the progenitor cell. In some cases, *S. pombe* can asymmetrically divide damaged proteins to inclusions (*Coelho et al., 2014*), but the mechanism is not as efficient as *S. cerevisiae*'s exclusion of

aggregates from buds. Due to this, it may be more important for *S. pombe* to destroy aggregates in the vacuole. The structure of the vacuoles is also fundamentally different in the two species, as *S. cerevisiae* tends to have one large vacuole per cell, while *S. pombe* has many small, distributed vacuoles. Additionally, the mechanisms determining the site of inclusions in *S. pombe* remain to be elucidated. Interestingly, *S. pombe wtf4*, the homolog of the *S. kambucha wtf4* allele used in this study, was previously shown to localize both inside the vacuole and as large cytoplasmic inclusions when overexpressed in *S. pombe* vegetative growth (*Matsuyama et al., 2006*). This suggests that some Wtf proteins can be recruited to both vacuoles and inclusions in *S. pombe*.

When we disrupted the ability of *S. cerevisiae* cells to transport the Wtf4[antidote] aggregates with the *vps1Δ* mutation, we found that the Wtf4[antidote] aggregates were distributed and more toxic than in wild-type cells. This is consistent with the idea that a key feature of Wtf4 protein toxicity relies on the aggregates being widely dispersed in the cytoplasm. When Wtf4[poison] and Wtf4[antidote] are found together in wild-type cells, the proteins co-assemble into aggregates. The co-assembled aggregates then behave similarly to the Wtf4[antidote] aggregates and are trafficked into the vacuole (in *S. pombe* cells) or to or near the IPOD adjacent to the vacuole (in *S. cerevisiae* cells) where they cause limited toxicity. Also, like the Wtf4[antidote] aggregates, the toxicity of the Wtf4[poison]+Wtf4[antidote] co-assembled aggregates is greatly enhanced if aggregate transport to the vacuole is disrupted by mutations (e.g. *vps1Δ*).

Together, our observations suggest a mechanistic model for *wtf4* function. In this model, *wtf4* exploits protein aggregation control pathways to induce selective cell death. The Wtf4[poison] forms distributed toxic aggregates and the Wtf4[antidote] co-assembles with the Wtf4[poison] and neutralizes the aggregate's toxicity via trafficking to the vacuole (*Figure 6*). This mechanism is unlike the mechanism of any other meiotic driver described to date (*Grognet et al., 2014*; *Didion et al., 2015*; *Long et al., 2008*; *Dawe et al., 2018*; *Rhoades et al., 2019*; *Dalstra et al., 2005*; *Hammond et al., 2012*; *Vogan et al., 2019*; *Chen et al., 2008*; *Akera et al., 2017*; *Bauer et al., 2012*; *Pieper et al., 2018*; *Herrmann et al., 1999*; *Shen et al., 2017*; *Yu et al., 2018*; *Bauer et al., 2007*; *Wu et al., 1988*; *Xie et al., 2019*; *Kruger et al., 2019*; *Lin et al., 2018*; *Svedberg et al., 2020*), but there are very few mechanistically characterized killer meiotic drive systems (reviewed in *Bravo Núñez et al., 2018b*).

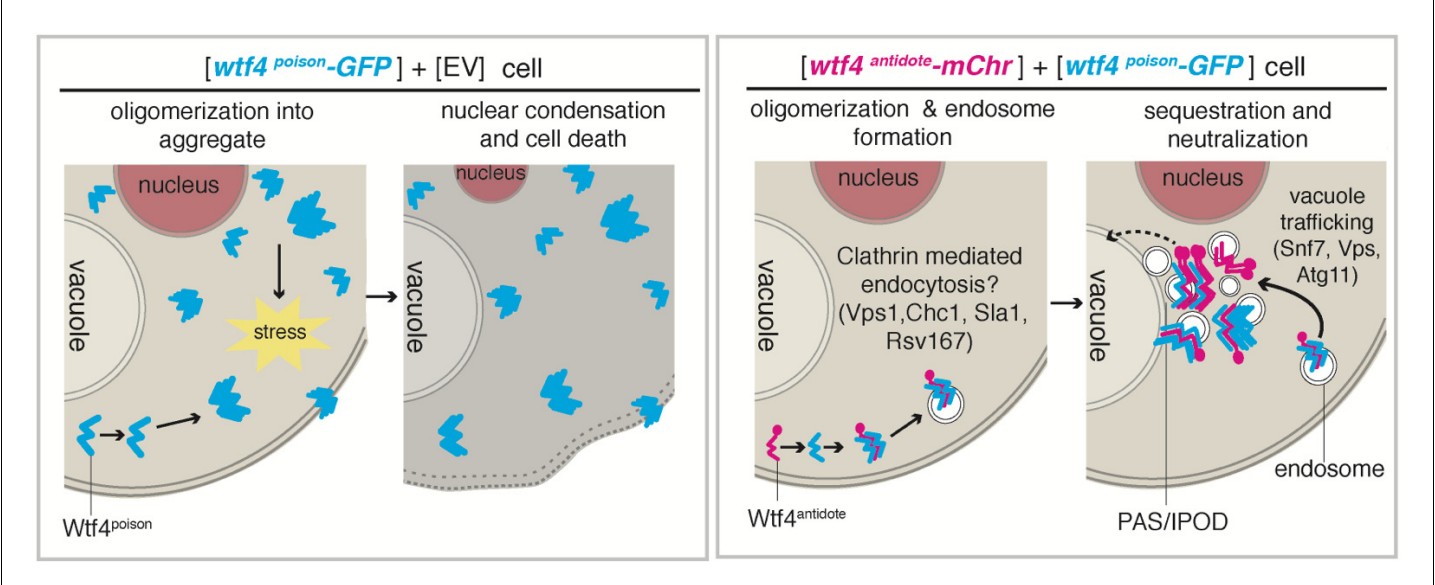

**Figure 6.** Model of Wtf4[poison] and Wtf4[antidote] mechanism in *S. cerevisiae*. Wtf4[poison] assembles into toxic aggregates that spread throughout the cell potentially causing stress. This stress could lead to nuclear condensation and ultimately cell death. Wtf4[antidote] co-assembles with Wtf4[poison], at least partially driven by shared sequences within the C-terminus. These assemblies are recruited to a compartment next to the vacuole and are also vesicle-associated. This sequestration neutralizes the Wtf4[poison] toxicity, rescuing cell viability.

## A controlled protein aggregation model offers a solution to the *wtf* diversity paradox

This study focused on the *wtf4* meiotic driver. There is, however, an incredibly diverse array of *wtf* genes that cause meiotic drive. For example, the poison protein encoded by *wtf35* (from the FY29033 isolate) shares less than 23% amino acid identity with Wtf4poison (*Bravo Núñez et al., 2020a*). Despite that extreme divergence, both genes cause essentially the same phenotype in *S. pombe* meiosis: drive of the gene into >90% of the progeny of a heterozygote (*Bravo Núñez et al., 2020a*). The conserved protein aggregation model offers an explanation for how such a diverse array of proteins can cause the same phenotype. Under our model, the mechanism of the Wtfpoison proteins is dependent upon their aggregation propensity. Presumably, the evolution of a protein that must self-aggregate could be less constrained than the evolution of a protein that must maintain a specific enzymatic activity or interaction partner.

Exon 1 of the *wtf4* driver encodes the antidote-specific residues (45 amino acids) that facilitate the recruitment of Wtf4 aggregates to the vacuole in *S. pombe* (or the IPOD in *S. cerevisiae*). This antidote-specific function likely relies on unidentified interacting partners (perhaps amongst our screen hits). This specific functional requirement could explain the greater conservation of exon 1-encoded residues amongst *bona fide wtf* drivers (68–100% amino acid identity) compared to the conservation amongst the remaining exons (30–90% amino acid identity) (*Bravo Núñez et al., 2020a*).

Importantly, our model also suggests that aggregate management may be a major feature of gametogenesis in *S. pombe*. The number of *wtf* genes varies between different isolates, but most have 30 or more *wtf* genes (*Eickbush et al., 2019*). Four of these genes are widely diverged from *wtf4* and are either not expressed in gametogenesis or their proteins exhibit distinct cellular localization from known Wtfpoison and Wtfantidote proteins (*Bravo Núñez et al., 2020a*). The rest of the genes that have been tested are similar to *wtf4* in expression and localization (*Bravo Núñez et al., 2018a*; *Bravo Núñez et al., 2020a*). It is not clear how many *wtf* drivers are expressed in a given cell, but they all appear to be transcribed at some level (*Eickbush et al., 2019*; *Kuang et al., 2017*). It will be interesting to explore the direct and indirect impacts of these Wtf proteins on *S. pombe* gametogenesis.

## Additional cellular factors are required to neutralize toxicity of Wtf4 proteins

The genetic screen presented in this work identified a number of factors required for cell viability in *S. cerevisiae* cells expressing Wtf4poison and Wtf4antidote. Many of these genes informed our model for Wtf4 protein function and therefore fit nicely within our proposed model. For example, our screen implicated genes involved in the Cytoplasm-to-Vacuole Targeting (CVT) pathway as necessary for survival of the Wtf4 proteins. This pathway has been previously implicated in aggregate management (*Kumar et al., 2016*; *Kumar et al., 2017*).

Not all of our screen hits, however, are in genes or pathways with annotated roles that clearly fit our model. Some of the genes have no annotated functions. It is possible that at least some of these genes are not directly involved in aggregate management, but the mutants are especially sensitive to the stresses imposed by Wtf4 aggregates. It is also possible that some of the genes do have roles in mitigating the effects of toxic aggregates. Indeed, in deletions of some genes with unknown functions, we saw distributed Wtf4 aggregates, suggesting these unknown proteins could play a role in sequestration of aggregates. Interestingly, other hits are in well-studied genes, such as multiple acetyltransferases and various kinetochore proteins. Future analysis of these hits will be essential to refine or to potentially reject our current model.

## Insight into protein cellular response to aggregates via studying meiotic drive

Studying how parasites manipulate their hosts can uncover unexpected insights on the host's biology. For example, studies of the mouse *t*-haplotype meiotic driver revealed that gene expression in spermatids can create sperm-autonomous phenotypes, even though spermatids are connected by intercellular bridges (*Herrmann et al., 1999*). Under our model, a fine line exists between protein aggregates that cells can manage (i.e. Wtf4antidote) and lethal aggregates that are not effectively

managed (i.e. Wtf4[poison]). We propose that the *wtf4* meiotic driver has exploited this feature for its own selfish advantage. Future studies can now exploit the Wtf4 proteins to learn about protein aggregate toxicity and cellular aggregate management strategies.

## Materials and methods

We confirmed all the vectors we generated (described below) via Sanger sequencing.

### Generation of tagged *wtf4* alleles for expression in *S. pombe* gametogenesis

#### Generation of a vector containing *wtf4*[antidote]-*mCherry* expressed from the endogenous promoter

We amplified the beginning of *wtf4* (including the endogenous promoter) from pSZB260 (described below) using oligos 688+719. The rest of the *wtf4*[antidote]-*mCherry* sequence was amplified (using oligos 605+1751) from pSZB708 (described below). The *ADH1* transcriptional terminator sequence was amplified from pSZB203 (*Nuckolls et al., 2017*) using oligos 1750+634. We then used overlap PCR (using oligos 688+634) to combine the three pieces. We then cloned the complete *wtf4*[antidote]-*mCherry* cassette into the SacI site of pSZB331 (*Bravo Núñez et al., 2020a*) to generate pSZB891.

#### Generation of a vector containing *wtf4*[antidote]-*GFP* expressed from the endogenous promoter

We amplified the upstream sequence and the beginning of the *wtf4* allele from pSZB203 (*Nuckolls et al., 2017*) using oligos 620+736. We amplified the rest of the *wtf4-GFP* sequence (with an *ADH1* transcriptional terminator) from pSZB203 using oligos 735+634. Oligos 736 and 735 introduced mutations that interrupt the Wtf4[poison] start site within intron 1. We then used overlap PCR with oligos 620+634 to unite the two pieces. We digested the complete *wtf4*[antidote]-GFP cassette with SacI site and cloned it into the SacI site of pSZB188 (*Nuckolls et al., 2017*) to generate pSZB260.

#### Generation of a vector containing the predicted *wtf4*[antidote] coding sequence expressed from the endogenous promoter

We amplified the *wtf4* coding sequence in three pieces. We amplified the promoter with oligos 633+604 using SZY13 DNA as a template. We amplified the coding sequence from a gBlock DNA fragment (Integrated DNA Technologies, Inc, Coralville) using oligos 605+614. We amplified the sequence downstream of *wtf4* using oligos 613+635 and SZY13 genomic DNA as a template. We then stitched the three pieces together using overlap PCR with oligos 633+635. We then digested the product with SacI and ligated the cassette into SacI-digested pSZB188 (*Nuckolls et al., 2017*) to generate pSZB199. Intron five was predicted wrong, so there is a mutation at the C-terminus. Within this study, this plasmid was only used to build other plasmids, and when used in subsequent steps, we repaired the C-terminal mutation with the PCR oligos.

### *S. pombe* Z$_3$EV β-estradiol inducible system

Z$_3$EV promoter system is a titratable inducible promoter system (*Ohira et al., 2017*). The system requires the Z$_3$EV transcription factor and a Z$_3$EV-responsive promoter (Z$_3$EVpr). β-estradiol induces nuclear import of the Z$_3$EV protein; therefore, genes placed immediately downstream of Z$_3$EVpr in a strain expressing Z$_3$EV become expressed upon β-estradiol addition to the media.

#### Background strain construction

To integrate the Z$_3$EV transcription factor at the *leu1* locus of *S. pombe*, we digested plasmid pFS461 (Addgene #89064, *Ohira et al., 2017*) with XhoI and transformed it into the yeast strain SZY643 (selecting for Leu+) via standard lithium acetate protocol (*Gietz et al., 1995*). This generated the yeast strain SZY2690, into which we transformed all of the proteins with Z$_3$EV promoters (see below).

## Generation of a strain that expresses *wtf4^antidote^-mCherry* under the control of a β-estradiol inducible promoter

We amplified the Z$_3$EVpr from pFS478 (Addgene #89066, *Ohira et al., 2017*) using oligos 1734 +1735. We then amplified the *wtf4^antidote^-mCherry* sequence (with an *ADH1* transcriptional terminator) from pSZB891 (described above) using oligos 1738+634. We used overlap PCR to add the Z$_3$EV promoter piece to the *wtf4^antidote^-mCherry* piece using oligos 1734+634. We then digested this cassette with SacI and ligated it into the SacI site of pSZB322 (*Bravo Núñez et al., 2018a*), a *lys4* integrating vector with a *hphMX6* cassette, to create pSZB892. We cut pSZB892 with KpnI and integrated into the *lys4* locus of SZY2690 to create SZY2740.

## Generation of a strain that expresses *wtf4^antidote^-mCherry* and *wtf4^poison^-GFP* under the control of β-estradiol inducible promoters

To create an estradiol inducible Wtf4^poison^-GFP vector, we amplified the Z$_3$EVpr on pSZB892 (see above) using oligos 1734+2068. We amplified the *wtf4^poison^-GFP* (with an *ADH1* transcriptional terminator) from pSZB203 (*Nuckolls et al., 2017*) using oligos 2069+634. We then completed overlap PCR (using oligos 1734+634) on the two pieces. We then digested the completed *wtf4^poison^-GFP* cassette with SacI and ligated it into the SacI site of pSZB331 (*Bravo Núñez et al., 2020a*) to create pSZB975. We cut pSZB975 with KpnI and integrated into the *ura4* locus of SZY2740 to generate SZY2888.

For the above transformations, we used high-efficiency, lithium acetate transformation protocol (*Gietz et al., 1995*) to integrate the vectors, selecting first for drug resistance and then screening for the relevant auxotrophy.

### Induction of Wtf proteins

For imaging cells (*Figure 1E and F*, *Figure 1—figure supplement 3*), we created 5 mL saturated overnight cultures in rich YEL broth (0.5% yeast extract, 3% glucose, 250 mg/L of adenine, leucine, lysine, histidine, and uracil) supplemented with 100 μg/mL G418 and Hygromycin B (to select against pop-outs of the *lys4* and *ura4* integrating plasmids described above). The next day, we diluted 1 mL of each saturated culture into 4 mL of fresh YEL+G418+HYG media. We then added β-estradiol (from VWR, #AAAL03801-03) to a final concentration of 100 nM and shook the cultures at 32°C for 4 hr. We then used these induced cultures for imaging (see below for microscopy details).

### *S. cerevisiae* LexA-ER-AD β-estradiol inducible system

The LexA-ER-AD system (*Ottoz et al., 2014*) utilizes a heterologous transcription factor containing a LexA DNA-binding protein, the human estrogen receptor (ER) and an activation domain (AD). β-estradiol binds the ER and tightly regulates the activity of the LexA-ER-AD transcription factor. The LexA DNA-binding domain recognizes *lexA* boxes in the target promoter.

### Background strain construction

To integrate the PACT1-LexA-ER-haB42 transcription factor into the *his3-11,15* locus of *S. cerevisiae*, we digested plasmid FRP718 (Addgene #58431, *Ottoz et al., 2014*) with NheI and transformed it into the yeast strain SLJ769 via standard lithium acetate protocol (*Gietz et al., 1995*) selecting for His+ cells. This generated SZY1637, the β-estradiol inducible PACT1-LexA-ER-haB42 transcription factor strain, into which we transformed all the plasmids carrying genes under the control of LexA box containing promoters (LexApr) (see below). We generally used the 'Sleazy' transformation protocol (incubate 240 μL 50% PEG3500 + 36 μL 1M Lithium Acetate + 50 μL boiled salmon sperm + 1–5 μL DNA + a match head amount of yeast overnight at 30°C; modified from *Elble, 1992*) to introduce plasmids into *S. cerevisiae*. We selected transformants on Synthetic Complete (SC) media (6.7 g/L yeast nitrogen base without amino acids and with ammonium sulfate, 2% agar, 1X amino acid mix, 2% glucose) lacking appropriate amino acids for selection of the transformed plasmids.

## Generation of a vector containing *wtf4^poison^-GFP* under the control of a β-estradiol inducible promoter

We amplified the LexApr from on FRP1642 (Addgene #58442, *Ottoz et al., 2014*) using oligos 1195 +1240. We cloned the promoter into the KpnI/XhoI sites of pSZB464 (see below) to create pSZB585.

## Generation of empty vectors containing only the β-estradiol inducible promoter

We amplified the LexApr from FRP1642 (Addgene #58442, *Ottoz et al., 2014*) using oligos 1195 +1240. We cloned that promoter into KpnI+XhoI-digested pRS314 (ARS CEN *TRP1* vector, *Sikorski and Hieter, 1989*) to generate pSZB668. We also cloned it into the KpnI+XhoI site of pRS316 (ARS CEN *URA3* vector, *Sikorski and Hieter, 1989*) to generate pSZB670.

## Generation of a vector containing *wtf4^antidote^-mCherry* under the control of a β-estradiol inducible promoter

We first amplified *wtf4^antidote^* (using oligos 1402+1401), *mCherry* (using oligos 1400+1399), and the *CYC1* transcriptional terminator (using oligos 1398+964) individually, using pSZB700 (see below) as the PCR template in all three reactions. We then used overlap PCR (using oligos 1402+964) to unite the three pieces into the *wtf4^antidote^-mCherry* cassette. We then digested the LexApr out of pSZB668 (with KpnI+XhoI) and ligated it, along with the *wtf4^antidote^-mCherry* cassette (digested with XhoI+BamHI), into KpnI+BamHI-digested pRS314 (*Sikorski and Hieter, 1989*) to generate pSZB708.

## Generation of a vector containing *mCherry-wtf4^antidote^* under the control of a β-estradiol inducible promoter

We amplified the *mCherry-wtf4^antidote^* sequence from pSZB248 (*Nuckolls et al., 2017*) using oligos 1066+604 and amplified the C-terminus of *wtf4^antidote^* plus the *CYC1* terminator from pSZB497 (see below) using oligos 1065+964. We then used overlap PCR (using oligos 1326+964) to join the two pieces. We digested the LexApr out of pSZB668 using KpnI and XhoI, and ligated that promoter, along with the XhoI-BamHI-digested *mCherry-wtf4^antidote^* cassette made by overlap PCR into KpnI-BamHI-digested pRS314 (*Sikorski and Hieter, 1989*) to generate pSZB700.

## Generation of a vector containing *wtf4^antidote^* under the control of a β-estradiol inducible promoter

We digested the galactose inducible promoter out of pSZB497 (see below) with KpnI and XhoI. We amplified the LexApr (using oligos 1195+1240) from FRP1642 (Addgene #58442, *Ottoz et al., 2014*), digested with KpnI+XhoI, and ligated it into KpnI-XhoI-digested pSZB497 to generate pSZB589.

## Induction of Wtf4 proteins with β-estradiol

For imaging, we grew 5 mL saturated overnight cultures in SC -His -Ura -Trp (without agar). The next day, we diluted 1 mL of the saturated culture into 4 mLs of media of the same type. We then added β-estradiol to a final concentration of 500 nM and shook the cultures at 30°C to induce. Cells were induced for 4 hr and then imaged at one or multiple timepoints, depending on the experiment. For spot assays, we diluted saturated cultures to an OD of ~1, then serial diluted ($10^0$, $10^{-1}$, $10^{-2}$, $10^{-3}$, $10^{-4}$) in a 96-well plate. We then spotted 10 uL of each dilution onto SC -His -Ura -Trp media and the same media with 500 nM β-estradiol. We grew the plates 2 to 3 days at 32°C and imaged on a SpImager (S and P Robotics).

### *S. cerevisiae* galactose inducible system

## Generation of a vector containing *wtf4^antidote^* under the control of a galactose inducible promoter

We amplified the beginning of the *wtf4^antidote^* coding sequence from pSZB388 (see below) using oligos 1065+678 and amplified the rest of *wtf4^antidote^* (and the *CYC1* transcriptional terminator) from pSZB392 (see below) using oligos 679+964. We then used overlap PCR with oligos 1065+964 to join

the two pieces. We then digested the complete *wtf4^antidote^* cassette with XhoI and BamHI and ligated them into XhoI+BamHI-digested pDK20 (*DasGupta et al., 1998*) to generate pSZB497.

## Generation of a vector containing *wtf4^poison^*-GFP under the control of a galactose inducible promoter

We first amplified *wtf4^poison^* followed by a *CYC1* terminator from pSZB388 (see below) using oligos 963+964. We then digested the PCR product with XhoI and BamHI and ligated it into XhoI+BamHI-digested pDK20 (*DasGupta et al., 1998*). This created pSZB392, a *URA3* integrating vector with *wtf4^poison^* under the control of a *GAL* promoter. We then amplified *wtf4^poison^* (including the *GAL* promoter) from pSZB392 with oligos 1045+606 and amplified GFP followed by an *ADH1* transcriptional terminator from pSZB203 (*Nuckolls et al., 2017*) using oligos 998+1040. We then stitched those two PCRs together using overlap PCR (amplifying with oligos 1040+1045). Finally, we digested the PCR product with KpnI and BamHI and cloned it into KpnI+BamHI-digested pRS316 (*Sikorski and Hieter, 1989*) to generate pSZB464 and into KpnI+BamHI-digested pRS314 (*Sikorski and Hieter, 1989*) to generate pSZB463.

## Generation of a vector containing *wtf4^antidote^*-mCherry under the control of a galactose inducible promoter

We amplified the *wtf4^antidote^* sequence with the galactose inducible promoter from pSZB497 (see above) using oligos 1929+997 and the *wtf4^antidote^*-mCherry sequence (with a *CYC1* terminator) from pSZB708 (see above) using oligos 1072+964. We then used overlap PCR using oligos 1929+964 to combine the two pieces. We then digested the complete *wtf4^antidote^*-mCherry cassette with BamHI and ligated it into BamHI-digested PRS315 (*Sikorski and Hieter, 1989*) to generate pSZB1005.

## Generation of a vector containing *wtf4^poison^*coding sequence

We amplified the *wtf4^poison^* coding sequence from pSZB199 (see above) using oligos 916+926. We then digested the PCR product with SfiI and cloned into SfiI-digested pBT3-STE (P03233DS from DUALsystems Biotech) to generate pSZB388. Within this study, this plasmid was only used to build other plasmids.

## Induction of Wtf4 proteins with galactose

For imaging, we grew 5 mL saturated overnight cultures in SC media lacking appropriate amino acids for selection of the plasmids. The next day, we pelleted the cultures, resuspended in YP raffinose media, and grew overnight. The next day, we diluted 1 mL of the saturated raffinose culture into 4 mL of SC galactose media lacking amino acids for selection of plasmids. We then added β-estradiol to a final concentration of 500 nM and shook the cultures at 30°C for four hours to create induced samples. For spot assays, we diluted saturated cultures to an OD of ~1, then serial diluted ($10^0$, $10^{-1}$, $10^{-2}$, $10^{-3}$, $10^{-4}$) in a 96-well plate. We then spotted 10 µL of each dilution onto both SC media (lacking amino acids appropriate for selection of the plasmids) and SC galactose media lacking the same amino acids. We grew the plates 2 to 3 days at 32°C and imaged them on a SpImager (S and P Robotics).

## Construction of the ER marker in *S. pombe*

To create the Sec63-YFP strain, we PCR amplified the C-terminus of *sec63* (using oligos 939+941) and the sequence downstream of *sec63* (using oligos 945+946) using SZY643 as a template. We also amplified a *YFP-HIS3* cassette from pYM41 (*Janke et al., 2004*) using oligos 944+943. We then used overlap PCR (using oligos 939+943) to unite those three PCR products. We then transformed this PCR product into GP1163 with standard lithium acetate protocol (*Gietz et al., 1995*) (selecting for His+) to integrate the tagged *sec63-YFP* at its endogenous locus to generate SZY1277. We confirmed the strain via PCR using oligos 2037+2038.

## Generation of the IPOD marker for expression in *S. cerevisiae*

### Generation of a vector containing *RNQ1-mCardinal* under the control of a β-estradiol inducible promoter

We amplified the LexApr from pSZB708 using oligos 1835+1834, the *RNQ1* sequence from pDK412 (*Kryndushkin et al., 2012*) using oligos 1833+1832 and the mCardinal-CYC1 terminator from V08_mC using oligos 1831+964. We then used overlap PCR (using oligos 1835+964) to stitch the three pieces together. We then digested the cassette with BamHI and ligated it into BamHI-digested pRS315 (*Sikorski and Hieter, 1989*) to generate pSZB942.

## Generation of chaperone overexpression strains in *S. cerevisiae*

### Generation of strains carrying *HSP104* under the control of a galactose inducible promoter

We used Gateway cloning to insert a *HSP104* cassette (from RH027) into RH14173, a *CEN* vector with a galactose inducible promoter. This generated the *pGal-HSP104* vector, pSZB997, which we then 'sleazy' transformed (incubate 240 μL 50% PEG3500 + 36 μL 1M Lithium Acetate + 50 μL boiled salmon sperm + 1–5 μL DNA + a match head amount of yeast overnight at 30℃; modified from *Elble, 1992*) into SZY1821 and SZY1766 to generate SZY2958 and SZY2956, respectively.

### Generation of strains carrying other chaperones under the control of galactose inducible promoters

The other chaperone plasmids were gifts from the Si lab, which were originally obtained from Yeast ORF collection in BG1805 vector or HIP FLEXGene ORF collection in BY011 vector. In both vectors, the chaperones are cloned under the galactose-inducible yeast *GAL1* or *GAL10* promoter, respectively. These were 'sleazy' transformed into SZY1637 to create SZY3116-SZY3126.

## Construction of *vps1Δ S. cerevisiae* strain

We PCR amplified the *vps1Δ::kanMX* locus out of strain YKR001C from the haploid yeast knockout *MATa* collection (Open Biosystems) (using oligos 1850+1851) and transformed the PCR product into SLJ769 using high efficiency lithium acetate protocol (selecting for G418 resistance) to create strain SZY2539. We used PCR (using oligos 1712+1713) and sequenced the locus to confirm the deletion. To add the PACT1-LexA-ER-haB42 transcription factor, we digested FRP718 (Addgene #58431, *Ottoz et al., 2014*) with NheI and integrated it into SZY2539 at the *his3-11,15* locus (using standard lithium acetate protocol [*Gietz et al., 1995*] and selecting for His+) to create SZY2552.

## Generation of the *wtf4* exon 6 mutant alleles for expression in *S. pombe*

### Generation of a vector containing *wtf4\*-GFP* allele under the endogenous promoter

We introduced the mutation within exon 6 of *wtf4* using PCR (oligos 1280 and 1281 contain the desired mutation). We amplified the endogenous promoter and beginning of *wtf4* using oligos 688 +1280 and pSZB203 as a template. We amplified the rest of *wtf4* and the downstream sequence using oligos 1281+686 and pSZB203 as a template. We then used overlap PCR using oligos 688 +686 to join the two pieces. Next, we digested the PCR product with SacI and cloned it into the SacI site of pSZB386 (*Bravo Núñez et al., 2018a*) to generate pSZB647. Within this study, this plasmid was only used to build other plasmids.

## Generation of the *wtf4* exon 6 mutant alleles for expression in *S. cerevisiae*

### Generation of a vector containing *wtf4$^{antidote}$\*-mCherry* under the control of a β-estradiol inducible promoter

We amplified the beginning of *wtf4$^{antidote}$* (using oligos 1402+1021) from pSZB700 (see above), the mutated section of *wtf4$^{antidote}$\** (using oligos 1072+997) from pSZB647 (see above) and mCherry-CYC1 terminator (using oligos 998+964) from pSZB708 (see above). We then stitched the three

pieces together (using 1402+964) to generate the complete *wtf4*$^{antidote}$*-mCherry* cassette and digested it with XhoI and BamHI. We also digested the LexApr from pSZB708 using KpnI and XhoI. We then cloned those digested pieces into KpnI-BamHI digested pRS314 to generate pSZB774.

### Generation of a vector containing *wtf4*$^{poison}$*-GFP* under the control of a β-estradiol inducible promoter

We amplified the beginning of *wtf4*$^{poison}$ (using oligos 1419+1021) from pSZB585 (see above). We amplified *wtf4*$^{poison}$* (using oligos 1072+997) from pSZB647 (see above) and GFP+ the *ADH1* terminator (using oligos 998+1040) from pSZB585 (see above). We then used overlap PCR (using oligos 1419+1040) to join the three pieces. We digested the cassette with XhoI and BamHI and also digested the LexApr from pSZB668 (see above) using KpnI and XhoI. We ligated the digested *wtf4*$^{poison}$*-GFP* cassette and the LexApr into KpnI-BamHI-digested-pSZB668 to generate pSZB786.

## Generation of the mEos3.1 tagged *wtf4* alleles

### Generation of vector containing *wtf4*$^{poison}$*-mEos3.1* under the control of a galactose inducible promoter

We used Golden Gate assembly (New England Biolabs) to insert the *wtf4*$^{poison}$ sequence that had been codon-optimized for *S. cerevisiae* (ordered from Addgene) into the BsaI-site of V08, a Gal-inducible vector with mEos3.1 and a rigid structure linker made up of a quadruple repeat of amino acids EAAAR [4x(EAAAR)] (*Khan et al., 2018*). This generated plasmid rhx1389.

### Generation of vector containing *wtf4*$^{antidote}$*-mEos3.1* under the control of a galactose inducible promoter

We used Gibson assembly (New England Biolabs) using oligos rh1282+rh1283 to insert a sequence that encodes the 45 amino acids of the codon-optimized *wtf4* exon1 into the Aar1-digested rhx1389 (see above) to create pSZB1120.

### Generation of a vector containing *wtf4*$^{poison}$*-mEos3.1* under the control of a β-estradiol inducible promoter

We amplified *wtf4*$^{poison}$*-mEos3.1* with a *CYC1* terminator sequence (using oligos 1466+964) from rhx1389 (see above) and digested with BamHI. We then ligated the cassette into the BamHI site of pSZB668 to generate pSZB732.

### Generation of a vector containing *wtf4*$^{antidote}$*-mEos3.1* under the control of a β-estradiol inducible promoter

We amplified the *wtf4*$^{antidote}$*-mEos3.1* with a *CYC1* transcriptional terminator (using oligos 1465+964) from pSZB1120 (see above) and digested with BamHI. We then ligated the cassette into the BamHI site of pSZB670 (see above) to generate pSZB756.

### Generation of a vector containing *wtf4*$^{antidote}$ under the control of β-estradiol inducible promoters

We digested the LexApro-*wtf4*$^{antidote}$*-CYC1* terminator construct out of pSZB589 (see above) and ligated it into pRS316 (*Sikorski and Hieter, 1989*) to create pSZB782.

## DAmFRET

We induced samples of SZY2072, SZY2070, SZY2159, and SZY2059, with β-estradiol as described above. We then aliquoted these induced samples into a 96-well plate. We then partially photoconverted the mEos3.1 protein by exposing the plate, while shaking at 800 RCF, to 405 nm illumination for 25 min using an OmniCure S1000 fitted with a 320–500 nm (violet) filter and a beam collimator (Exfo), positioned 45 cm above the plate. This exposure yielded a total photo dose of 16.875 J/cm$^2$. This photo dose reproducibly achieves the maximum fluorescence of the acceptor (red) form of mEos3.1 while minimizing photobleaching of the green form (*Khan et al., 2018*). For *Figure 2—figure supplement 4C*, we assayed the photoconverted samples on a Bio-Rad ZE5 cell analyzer with

high-throughput automation. We analyzed 20 μL of each sample to collect approximately 100,000 events per well. We excited the mEos3.1 donor (green form) with a 488 nm laser at 100 mW and collected with 525/35 nm and 593/52 nm bandpass filters, respectively. We excited the acceptor fluorochrome with a 561 nm laser at 50 mW and collected with a 589/15 nm bandpass filter. We performed manual compensation on-instrument at acquisition. We used DeNovo FCS Express for data analysis and visualization and calculated ratiometric FRET as FRET/acceptor signals.

## Fission yeast microscopy

For imaging during *S. pombe* gametogenesis (*Figure 1C, D and G*, *Figure 1—figure supplement 1B*, *Figure 1—figure supplement 2A and C*), we crossed the two haploid yeast strains to generate heterozygous diploids as previously described (*Nuckolls et al., 2017*). We placed the diploids on sporulation agar (SPA, 1% glucose, 7.3 mM $KH_2PO_4$, vitamins, agar) for 2–3 days. We then scraped the cells off of the SPA plates and onto slides coated with 0.2 mg/mL lectin (Sigma) for imaging (*Tomita and Cooper, 2007*).

For vegetatively growing samples (*Figure 1E and F*, *Figure 1—figure supplement 3B and D*), we induced gene expression with β-estradiol as described above. If we used vacuole staining, we took 1 mL of the induced culture, spun to pellet, and resuspended in 1 mL of 10 mM HEPES buffer, pH 7.4, containing 5% glucose with 100 μM CellTracker Blue CMAC (Component B; Invitrogen C2110). We incubated these cells at room temperature for 30 min. We then washed with YEL media and imaged. For imaging, we used the LSM-780 (Zeiss) microscope with a 40x C-Apochromat water-immersion objective (NA 1.2) in photon-counting channel mode. For GFP, we used 488 nm excitation and collected through a 491–552 bandpass filter. For mCherry, we used 561 nm excitation and collected through a 572 longpass filter. For YFP, we used 514 nm excitation and collected through a 500–589 nm bandpass filter. For CMAC, we used 405 nm excitation and collected through a 411–509 nm bandpass filter. Brightness and contrast are not the same for all images. We analyzed at least 20 cells for each strain and chose a representative image. For experiments assaying meiosis/gametogenesis, we used at least two independent progenitor diploids. For cells that were imaged during vegetative growth, we used at least three different starting cultures.

We carried out Pearson correlation analysis (*Adler and Parmryd, 2010*) as previously described (*Slaughter et al., 2013*). Briefly, we drew a segmented line (width of two pixels) throughout the spore, randomly covering as much of the spore as we could. We then used an in-house custom written plugin for ImageJ (https://imagej.nih.gov/ij/) to generate a two-color line profile. We calculated the Pearson correlation of the line profile with varying degrees of shifts in at least eight spores or six vegetatively growing cells per sample. We then combined and averaged the trajectories with standard error.

To quantify nuclear size, we calculated the full width at half maximum of the fluorescence intensity of RFP. We quantified 42 spores that inherited *wtf4-GFP* and 19 that did not, all from a *wtf4-GFP/ade6+* heterozygote after 2 days on SPA media. We excluded any nuclei that appeared to have already fragmented.

For the nuclear timelapse (*Figure 1—figure supplement 4*), we grew diploid cultures to saturation at 32°C overnight in YEL media. We then plated 100 μL of the cultures on a SPA plate, cut a circle punch of agar from the plate, and placed this punch upside down (cells facing down) in a 35 mm glass bottom poly-D-lysine coated dish (MatTek corporation). We placed grease around the edge of the MaTeK dish and a moist kim wipe inside to control for humidity. We then imaged the cells using the Nikon Ti Eclipse coupled to a Yokogawa CSU W1 Spinning Disk, using the 60x oil objective, acquiring images every ten minutes. Here, we excited RFP at 561 nm and collected its emission through a 605–70 nm bandpass filter.

For the gametogenesis timelapse (*Figure 1—figure supplement 1B*), we grew diploid cultures to saturation at 32°C overnight in YEL media. The next day, we diluted 100 μL of the saturated diploid culture into 5 mLs of PM media (20 mLs of 50x EMM salts, 20 ml 0.4 M $Na_2HPO_4$, 25 mL 20% $NH_4Cl$, 1 mL 1000x Vitamins, 100 μL 10,000x mineral stock solution, 3 g potassium hydrogen phthalate, 950 mL ddH$_2$O, 25 mL of sterile 40% glucose after autoclaving, supplemented with 250 mg/L uracil). We grew the PM culture overnight at 32°C. The next day, we spun to pellet and resuspended the pellet in PM-N media (20 mLs of 50x EMM Salts, 20 mL 0.4 M $Na_2HPO_4$, 1 mL 1000x Vitamins, 100 μL 10,000x mineral stock solution, 25 mL of sterile 40% glucose after autoclaving, supplemented with 250 mg/L uracil, volume up to 1 L with ddH$_2$O). We shook the PM-N cultures for 4 hr at 28°C.

Then, we took 100 µL of the PM-N culture and mixed it with 100 µL of lectin (Sigma). We took 150 µl of this mixture and added it to a 35 mm glass bottom poly-D-lysine coated dish (MatTek corporation). We waited 5 min to allow the cells to adhere. We then added 3 mL of fresh PM-N to the dish (protocol modified from *Klutstein et al., 2015*). We imaged using a Zeiss Observer.Z1 wide-field microscope with a 63x (1.4 NA) oil-immersion objective and collected the emission onto a Hamamatsu ORCA Flash 4.0 using µManager software. We acquired the mCherry with BP 530–585 nm excitation and LP 615 emission, using an FT 600 dichroic filter, acquiring images every 10 min.

## Budding yeast microscopy

For all budding yeast images except for the three experiments described below, we induced samples as described above and imaged on an LSM-780 (Zeiss) microscope, with a 40x C-Apochromat water-immersion objective (NA 1.2) in photon-counting channel mode. For GFP and mCherry, we used the same conditions as we did in *S. pombe.* For mCardinal, we used 633 nm excitation and collected through a 632–696 nm bandpass filter. Brightness and contrast are not the same for all images. We imaged at least 20 cells from at least three starting cultures and chose a representative image for each figure.

For imaging *vps1Δ* cells (*Figure 5*), IPOD cytology (*Figure 5—figure supplement 1A*), and the nuclear timelapse (*Figure 2—figure supplement 1B and C*), we placed samples in a Millipore Onix 2 Cellasic system to allow for a constant flow of media. We initiated flow of inducing media (SC with 500 nM b-estradiol for *vps1Δ* and SC galactose for the nuclear timelapse) and took images every 10 min. We used a Perkin Elmer Ultraview Vox spinning disc microscope with a Hamamatsu EMCCD (C9100-23B) with a 40x C-Apochromat water-immersion objective (NA 1.2). We collected GFP and mCherry with 488 and 561 nm excitation as above but collected GFP through a 525–550 nm bandpass filter and mCherry through a 615–670 nm bandpass filter. We had two independent starting cultures for the sample. We chose representative cells and timepoints. Brightness and contrast are not the same for all images.

To quantify nuclear size (*Figure 2—figure supplement 1D*), we fit each nucleus to a two-dimensional Gaussian function and found the full width at half maximum of the fluorescence intensity of RFP per cell. We quantified at the beginning of the timelapse (early) and 14 hr into the timelapse (late). We quantified 72 Wtf4$^{poison}$-GFP expressing cells and 65 wild-type cells at the early timepoint. We quantified 62 Wtf4$^{poison}$-GFP expressing cells and 79 wild-type cells at the later timepoint.

## Western analysis of Wtf4$^{poison}$ in *S. cerevisiae*

We grew overnight cultures of SZY1954 (cells carrying inducible Wtf4$^{poison}$-GFP, Wtf4$^{antidote}$-mCherry) and SZY1821 (empty vector (EV), untagged control) in SC -His -Ura -Trp (without agar). The next day, we diluted 10 mL of the saturated culture into 90 mL of media of the same type. We then added β-estradiol to a final concentration of 500 nM. We shook the cultures at 30°C to induce. After 2 and 5 hr of induction, we prepared whole-cell lysates as previously described (*Gerace and Moazed, 2014*), with a few exceptions. First, we used 1% tritonX-100 instead of NP40 in the lysis buffer. For bead beating, we used the FastPrep-24 5G bead beater with Lysing Matrix Y bead tubes (MP Biomedicals) and the *S. cerevisiae* fast prep program (40 s). We took both the supernatant and pellet protein samples for analysis. Samples were boiled for 1 min at 95°C before loading.

Next, we ran the proteins on NuPAGE 4–12% Bis-Tris Protein Gels (Invitrogen, NP0321) and then transferred to Trans-Blot Turbo Mini PVDF membranes (Bio-RAD #1704156). We stained the membranes with a monoclonal, α-GFP antibody (from cell signaling technology #2956) at 1:1000 and monoclonal, α-mCherry antibody (from EnCor Biotechnology Inc #MCA-1C151) at 1:1000, overnight at 4°C with agitation in Odyssey blocking buffer (TBS, from LI-COR biosciences). A secondary, α-rabbit antibody (800 cW) and secondary, α-mouse antibody (600 cW) was used for fluorescent visualization of the proteins. We imaged the blot on the Odyssey-CLx (LI-COR biosciences). Three independent starting cultures were analyzed to confirm results.

## Acceptor photobleaching FRET in *S. cerevisiae*

We carried out acceptor photobleaching FRET with β-estradiol induced (described above) SZY1954 (wild type) using a LSM-780 (Zeiss) microscope, with a 40x C-Apochromat water-immersion objective (NA 1.2) in photon-counting channel mode. For *vps1Δ* cells (SZY2570), we used a Perkin Elmer

Ultraview Vox spinning-disc microscope with a Hamamatsu EMCCD (C9100-23B) with 488 and 561 nm excitation. For both samples, we photobleached the acceptor (mCherry) with 561 nm excitation (for bleaching images, see *Figure 2—figure supplement 1F* for wild type and *Figure 5—figure supplement 5A* for *vps1Δ*). We analyzed 22 wild-type and 76 *vps1Δ* cells.

## Fluorescence recovery after photobleaching half-FRAP of Wtf4 aggregates

### In *S. cerevisiae*

We induced SZY1954 with β-estradiol as described above and mounted into a lectin-coated 35 mm glass bottom poly-D-lysine-coated dish (MatTek corporation) and imaged on a Perkin Elmer Ultraview Vox spinning disc with a Hamamatsu EMCCD (C9100-23B). We excited GFP with a 488 nm laser and collected its emission through a 100x alpha plan Apochromat objective (NA = 1.4) and a 525–50 nm bandpass filter. For each cell (n = 10), we bleached half of the visible aggregate. We then acquired recovery images every second for three minutes total time.

### In *S. pombe*

We placed SZY1142/SZY1049 heterozygous diploids on SPA plates for 2 days. We then scraped the sample off of the SPA plates into a 35 mm glass bottom poly-D-lysine-coated dish (MatTek corporation) and carried out half-FRAP as above (n = 10 spores), except that recovery images were then acquired for three minutes total time.

## Electron microscopy

We made 50 mL saturated overnight cultures of SZY1821, SZY1952, SZY1954, and SZY2731 in SC media lacking histidine, tryptophan, and uracil (to select for retention of the plasmids). The next day, we diluted 10 mL of the saturated cultures into 90 mL of the same media with 500 nM β-estradiol. We shook these cultures for four hours at 30°C, reaching log phase. We then pelleted the yeast cells by filtering and carried out high pressure freezing with the Leica ICE system (Leica Biosystems). We further processed the frozen cell pellets by freeze substitution (FS) using acetone containing 0.2% uranyl acetate (UA) and 2% $H_2O$ was used as FS medium. The FS program was −90° to −80° over 70 hr, −80° to −60° over 6 hr, −60° for 5 hr, −60° to −50° over 6 hr, and −50° to −20°C over 4 hr. After washing extensively with acetone, we then infiltrated, embedded and polymerized the samples into resin.

For Immuno-EM, we used HM-20 resin. We cut 60 nm sections with a Leica Ultra microtome (Leica UC-6) and picked up onto a carbon-coated 150 mesh nickel grid. The grids were labeled with anti-GFP primary antibody (a gift from M. Rout, Rockefeller University, New York, NY) and 12 nm colloidal gold goat anti-rabbit secondary antibody (Jackson Immuno Research Laboratories, Inc). After immuno labeling, we post-stained the samples with 1% UA for 3 min. We acquired images using a FEI Tecnai Biotwin electron microscope. For non-immuno-EM, we used Epon resin to better maintain morphology, but the rest of the procedure was the same. We analyzed the tomographs of at least 10 cells per condition.

For quantification purposes, we also completed array tomography. For array tomography, we cut 60 nm serial sections with a Leica Ultra microtome (UC-6) using an Ultra 35 Jumbo diamond knife (Diatome) and picked up on ITO coated coverslips using the ASH-100 Advanced Substrate Holder (RMC Boeckeler). We post-stained serial sections with Sato's triple lead stain for two minutes, 4% UA in 70% methanol for two minutes, and Sato's lead stain again for two minutes. The coverslips were coated with 5 nm of carbon and imaged in a Zeiss Merlin Gemini 2 SEM with 4QBSD detector at 10 kV and 700 pA using Atlas 5 Array Tomography software (Fibics). The obtained dataset was aligned with Midas of the IMOD software package (*Kremer et al., 1996*) and manually quantified. For better visualization, an image series of an individual yeast cell was cropped and further aligned with registration tools in ImageJ.

For model building, the segmentation was done based on intensity and known organelle structure with Microscopy Image Browser (*Belevich et al., 2016*) and with IMOD. We used Amira (Thermo Fisher Scientific) software for model rendering and visualization.

Further quantification of mitochondrial volumes was performed on selected cells after training a Unet (*Ronneberger et al., 2015*). Hand annotation of training data was performed in Fiji. A suite of

internally developed Fiji plugins, macros and CherryPy scripts called DeepFiji (see below) sent training data to a pair of in-house NVIDIA Tesla-equipped deep learning machines running Tensorflow. Representative cells were selected, and segmented images inferred using the same macros and deep learning machines before being aligned using a StackReg variant. Mitochondrial volumes were quantified in Fiji using the 3D Segmentation tools.

## DeepFiji training

DeepFiji is a suite of macros and plugins in Fiji, Python, and CherryPy (a Python web framework) that enable end users on any machine with a reasonable amount of RAM to request deep learning training and inference on a remote deep learning box as long as both machines have access to a shared file system.

First, a user selects example sub-images that span the realm of potential objects, background levels and signal levels. Manual annotations are made using Fiji's Region of Interest (ROI) tools and manager, and individual ROI files are saved for each image (in our case the ROIs were each individual cell and the total of the mitochondria inside). A user chooses a small subset of the annotated image/ROI pairs to be used as a validation set, while the remainder becomes the training set. For each image, two binary channels are added from the associated ROIs: a mask channel and an outline channel. The mask channel has all pixels contained within an ROI painted true, while the outline channel only paints true the pixels that were in the ROI's outline. As the training network expects standard image sizes and runs more efficiently with smaller images, the macro next makes image stacks for both annotated training and validation images that break the original images up into 512 × 512 sub-images with 50% overlap between sub-images. For the training set, it also applies a series of random rotations and translations to help the neural network generalize. Both the validation and new training images are saved to a shared file system and the deep learning boxes are notified to begin processing through a call to a webserver running on the box. The sub-image size, file location, and other parameters are configurable by the user at the time of running.

For our in-house system, the deep learning boxes are running ubuntu with NVIDIA Tesla boards and configured with Tensorflow 1.13.1 and CUDA 10.0. CherryPy is configured to listen for web calls on each and, once initiated from a user, begins processing files from the selected directory by calling trainer.py. The trainer first finds the standard deviation (STDEV) and mean (MEAN) of the non-zero pixel intensities and stores those values. The training, validation, and all future inference sets will be processed by (Intensity-MEAN)/STDEV+0.5 first to keep numbers roughly between 0 and 1. The model used is a modified Unet(ref): convolutional layers are alternated with max pool layers, doubling the channel depth at each layer while halving the resolution. The final layer is 32 × 32×512. At each convolutional layer, the image is passed through a leaky rectified linear unit. Convolutional up-sampling brings the image back to its original resolution where it is passed through a tanh() function and the mean square error is calculated with respect to the ground truth image for back-propagation. Every 100th iteration of the training is applied to the validation set and the images are visualized using TensorBoard (which opens automatically on the user's host computer). Training proceeds with the learning rate adjusting over time, until 2200 iterations have passed at which point most networks have either converged or never will.

Once training is completed, and a reasonable iteration point is found in TensorBoard, the user can run Inferer.ijm in Fiji on their host machine to apply their model to a new dataset. Inferer will similarly parse images into sub-images and contact a deep learning box to initiate processing. Once processing is complete the user can run a second script to blank out border regions and de-window their images. The output outline and mask channels are ranged from 0 to 1 and represent probabilities. Typically, thresholding pixel values above 0.5 in the mask channel will suffice for finding objects of interest. However, in cases with frequent object touching, one can subtract the outline probability from the mask.

If consistent mistakes are found in the inferred data, the user can annotate them properly using Fiji and use Retrain.ijm to retrain the network using the new data together with the old to generate a new training set. Retraining starts from the original model so that it does not have to relearn from scratch.

Plugins necessary to run in Fiji are available from the Stowers update site within Fiji. Macros, python code, and CherryPy configurations are available at https://github.com/cwood1967/DeepFiji.

## Genetic screen for suppressors of Wtf4$^{antidote}$ function in *S. cerevisiae*

### Screen design

Two separate plasmids carrying galactose inducible Wtf4 proteins [Wtf4$^{poison}$-GFP (pSZB464) and Wtf4$^{antidote}$-mCherry (pSZB1005)] were transformed into the *MATa* haploid *S. cerevisiae* deletion collection (purchased from Open Biosystems) using lithium acetate protocol (*Gietz et al., 1995*). The transformed collection was then spot inoculated on SC and SC galactose media lacking leucine and uracil. As a control, a strain expressing the galactose inducible poison and antidote in a wild-type background was used for comparison. We grew the plates for three days at 30°C, imaged them using the SpImager (S and P Robotics), and manually scored growth. For the antidote-only screen, we transformed the galactose driven Wtf4$^{antidote}$-mCherry (pSZB1005) plasmid into the 106 hits from the first screen and scored them as mentioned above.

### Confirmation of hits

The initial screen identified 250 strains that grew poorly on inductive media. To confirm that this poor growth was due to the Wtf4 proteins and not due to the background strain being sick or a poor grower on galactose media in general, we completed a follow-up screen. We transformed the 250 strains we identified as 'poor growers' with empty [*URA3*] and [*LEU2*] vectors and assayed the strains as above to identify those that grew poorly on galactose media independent of *wtf4* gene expression. We found 106 strains that passed this secondary screen, which we then called hits. We imaged these 106 strains after a short galactose induction (~4 hr) to ensure we saw Wtf4 protein.

### Analysis

To look for enriched gene ontology terms in the hits from the screen, we used the PANTHER over-representation Test (*Thomas et al., 2003*; *Thomas et al., 2006*). The background list we used for the analysis was the list of *MATa* deletion collection strains that we successfully transformed with our plasmids of interest (n = 4793). We used Fisher's Exact test and corrected with false discovery rate. We imaged the cells as described above for Gal-inductions, but we added 80 mg/L adenine to the inducing media to circumvent any potential autofluorescence introduced by the adenine auxotrophy.

## Acknowledgements

We thank Dan Kaganovich, the other anonymous reviewers, as well as the members of the Zanders lab for their helpful comments on the paper. We thank Alejandro Rodríguez Gama, Julie Cooper, Li-Lin Du, Snezhana Oliferenko, Risa Mori, and Frank Shewmaker for strains or plasmids. We thank Kausik Si for valuable advice on the project, Sue Jaspersen for advice and reagents, and Xia Zhao for assistance in electron microscopy. We also thank Mark Miller for help with graphics. Original data underlying this manuscript can be accessed from the Stowers Original Data Repository at http://www.stowers.org/research/publications/libpb1505. This work was performed to fulfill, in part, requirements for NLN's thesis research in the Graduate School of the Stowers Institute for Medical Research. This work was supported by The Stowers Institute for Medical Research (SEZ, RH); the Searle Award (SEZ); National Institutes of Health (NIH) R00GM114436 and DP2GM132936 (SEZ); NIH Director's Early Independence Award DP5-OD009152 (RH); National Cancer Institute of the NIH under award number F99CA234523 (MABN). Eunice Kennedy Shriver National Institute of Child Health and Human Development of the NIH under Award Number F31HD097974 (NLN); The funders had no role in study design, data collection and analysis, or manuscript preparation. The content is solely the responsibility of the authors and does not necessarily represent the official views of the funders.

## Additional information

**Competing interests**

Nicole L Nuckolls, María Angélica Bravo Núñez: Inventor on patent application based on wtf killers. Patent application 834 serial 62/491,107'. Sarah E Zanders: Inventor on patent application based on

wtf killers. Patent application 834 serial 62/491,107. The other authors declare that no competing interests exist.

## Funding

| Funder | Grant reference number | Author |
| --- | --- | --- |
| Stowers Institute for Medical Research | | Randal Halfmann<br>Sarah E Zanders |
| Kinship Foundation | Searle Scholars Award | Sarah E Zanders |
| National Institutes of Health | R00GM114436 | Sarah E Zanders |
| National Institutes of Health | DP2GM132936 | Sarah E Zanders |
| National Institutes of Health | DP5OD009152 | Randal Halfmann |
| National Institutes of Health | F99CA234523 | María Angélica Bravo Núñez |
| National Institutes of Health | F31HD097974 | Nicole L Nuckolls |

The funders had no role in study design, data collection and interpretation, or the decision to submit the work for publication.

## Author contributions

Nicole L Nuckolls, Conceptualization, Data curation, Formal analysis, Funding acquisition, Validation, Investigation, Visualization, Methodology, Writing - original draft, Writing - review and editing; Anthony C Mok, Data curation, Formal analysis, Validation, Investigation, Writing - review and editing; Jeffrey J Lange, Kexi Yi, Data curation, Formal analysis, Validation, Investigation, Visualization, Methodology, Writing - review and editing; Tejbir S Kandola, Scott McCroskey, Data curation, Formal analysis, Validation, Investigation, Methodology, Writing - review and editing; Andrew M Hunn, Julia L Snyder, María Angélica Bravo Núñez, Formal analysis, Validation, Investigation, Methodology, Writing - review and editing; Melainia McClain, Formal analysis, Investigation, Visualization, Methodology, Writing - review and editing; Sean A McKinney, Formal analysis, Investigation, Methodology, Writing - review and editing; Christopher Wood, Data curation, Formal analysis, Investigation, Methodology, Writing - review and editing; Randal Halfmann, Supervision, Methodology, Writing - review and editing; Sarah E Zanders, Conceptualization, Data curation, Formal analysis, Supervision, Funding acquisition, Visualization, Project administration, Writing - review and editing

## Author ORCIDs

María Angélica Bravo Núñez (iD) https://orcid.org/0000-0002-6554-8814
Sarah E Zanders (iD) https://orcid.org/0000-0003-1867-986X

## Decision letter and Author response

Decision letter https://doi.org/10.7554/eLife.55694.sa1
Author response https://doi.org/10.7554/eLife.55694.sa2

# Additional files

## Supplementary files

• Supplementary file 1. Yeast strains. Column 1 is the name of strain used, while column 2 refers to the species of the yeast (Sp = S. pombe; Sc = S. cerevisiae). Columns 3 lists the reference for the yeast strain. If it was made in this study, we also detail how the strain was made in column 5. Column 6 lists which figures the yeast strain was used in.

• Supplementary file 2. Plasmids used in this study. Column 1 is the name of plasmid used, while column 2 describes the content of the plasmid. Columns 3 lists the reference for each plasmid.

• Supplementary file 3. Oligos used in this study. Column 1 lists the oligo numbers and column 2 provides the sequence of the oligo. Columns 3 gives a short description of each oligo.

• Transparent reporting form

## Data availability

All data generated or analysed during this study are included in the manuscript and supporting files. A source data file has been provided for Figure 5. All original data is available at the Stowers original data repository (ftp://odr.stowers.org/LIBPB-1505).

The following dataset was generated:

| Author(s) | Year | Dataset title | Dataset URL | Database and Identifier |
|---|---|---|---|---|
| Nuckolls NL, Mok AC, Lange JL, Yi K, Kandola TS, Hunn AM, McCroskey S, Snyder JL, Núñez MAB, McClain M, McKinney SA, Wood C, Halfmann R, Zanders SE | 2020 | The *wtf4* meiotic driver utilizes controlled protein aggregation to generate selective cell death | ftp://odr.stowers.org/LIBPB-1505 | Stowers Original Data Repository, LIBPB-1505 |

The following previously published dataset was used:

| Author(s) | Year | Dataset title | Dataset URL | Database and Identifier |
|---|---|---|---|---|
| Matsuyama A, Arai R, Yashiroda Y, Shirai A, Kamata A, Sekido S, Kobayashi Y, Hashimoto A, Hamamoto M, Hiraoka Y, Horinouchi S, Yoshida M | 2006 | ORFeome cloning and global analysis of protein localization in the fission yeast Schizosaccharomyces pombe. | https://www.pombase.org/reference/PMID:16823372 | PomBase.org, 16823372 |

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
