## [Decision Letter]

**Acceptance summary:**

In this manuscript, Nuckolls and co-authors investigated how the meiotic driver, wtf4, functions on a molecular level in distantly related yeasts to exert its function as a parasitic gene locus. They show that the wtf gene utilizes a poison-antidote mechanism whereby the wtf4 gene products, Wtf4poison and Wtf4antidote, exploit differential aggregation strategies to ultimately achieve their function. The study makes an unexpected connection between the meiotic driver field and protein aggregation quality control mechanisms.

**Decision letter after peer review:**

Thank you for submitting your article "The wtf4 meiotic driver utilizes controlled protein aggregation to generate selective cell death" for consideration by *eLife*. Your article has been reviewed by three peer reviewers, one of whom is a member of our Board of Reviewing Editors, and the evaluation has been overseen by Suzanne Pfeffer as the Senior Editor. The following individual involved in review of your submission has agreed to reveal their identity: Dan Kaganovich (Reviewer #3).

The reviewers have discussed the reviews with one another and the Reviewing Editor has drafted this decision to help you prepare a revised submission.

Essential revisions:

1) The authors show that Wtf^poison^ forms aggregates, but there is no clear evidence as to a plausible toxic mechanism. While a detailed analysis of the toxicity mechanism is beyond the scope of the present article, the authors should test whether overexpression of chaperone such as Hsp104 or Sis1 prevents Wtf^poison^ aggregation and toxicity in yeast.

2) The authors demonstrate that coexpression of Wtf4^antidote^ with Wtf4^poison^ suppresses the deleterious and toxic effect exerted by Wtf4 poison through a mechanism, they argue, that requires sequestration of the toxic Wtf isoform into either a spatially confined compartment (IPOD or PAS – observed in *S. cerevisiae*) or directly inside the vacuole (*S. pombe*). However, it is not entirely clear whether varying mechanisms are at play regarding the suppression of toxicity in the different yeasts (is it just that spatial sequestration, and not vacuolar degradation, is sufficient for suppression of toxicity in *S. cerevisiae*?). A Western blot analysis should be performed over the course of an estradiol induction in both *S. pombe* and *S. cerevisiae* vegetative cells to confirm that the toxin is degraded in *pombe* but not in cerevisiae; this would also address questions regarding overall levels of the proteins. Because GFP or RFP are resistant to proteolysis, the use of the fluorescence signal as a proxy for the fusion proteins needs to be substantiated on Western blots that will distinguish the fusion protein from free GFP/RFP.

3) A clear demonstration that Wtf4^antidote^ physically interacts with Wtf4^poison^ would be important for understanding the mechanism of toxicity suppression. The authors use FRET to provide evidence that the proteins indeed physically interact. To further support this, can the authors biochemically demonstrate a physical interaction between these proteins, for example by co-immunoprecipitation or another method? Short of demonstrating a direct physical interaction, it would be helpful to know whether both proteins come down in a pelletable aggregate.

4) The colocalization between the Wtf4 protein aggregates and the IPOD marker Rnq1-mCardinal isn't overly convincing. Can the authors quantify the number of cells that exhibit this kind of distribution?

[Editors' note: further revisions were suggested prior to acceptance, as described below.]

Thank you for submitting your revised article "The wtf4 meiotic driver utilizes controlled protein aggregation to generate selective cell death" for consideration by *eLife*. Your article has been reviewed by two peer reviewers, and the evaluation has been overseen by a Reviewing Editor and Suzanne Pfeffer as the Senior Editor. The reviewers have opted to remain anonymous.

The reviewers have discussed the reviews with one another and the Reviewing Editor has drafted this decision to help you prepare a revised submission.

The reviewers commented positively on the revised manuscript and recommended publication when you are able to address a remaining concern of one of the reviewers:

"In response to concern #2 in the decision letter, authors have only performed western blots for *S. cerevisiae*, while blots for both *S. pombe* and *S. cerevisiae* were requested. The blots are somewhat marginal in quality and do not contain a free GFP sample to allow unambiguous identification of the free GFP that would be diagnostic of vacuolar degradation, nor does it look as if there is much signal at the expected 27 kDa region of the gel. This, combined with the prominence of anti-GFP reactive bands of unexpected sizes, suggest that the text in subsection “Wtf4^antidote^ promotes neutralization of Wtf4^poison^ via recruitment to vacuole-associated sites” should be modified to acknowledge the possibility of other degradation mechanisms." Please address this concern in full by providing the requested data.

---

## [Author Response]

Essential revisions:1) The authors show that Wtf^poison^ forms aggregates, but there is no clear evidence as to a plausible toxic mechanism. While a detailed analysis of the toxicity mechanism is beyond the scope of the present article, the authors should test whether overexpression of chaperone such as Hsp104 or Sis1 prevents Wtf^poison^ aggregation and toxicity in yeast.

Thank you for this suggestion. We tested this idea and the results are depicted in Figure 2—figure supplement 5 and discussed in subsection “Wtf4 poison and antidote proteins assemble into aggregates individually and together in budding yeast”. We tested six chaperones, including Hsp104 and Sis1 and saw that none suppressed toxicity of Wtf4^poison^.

2) The authors demonstrate that coexpression of Wtf ^antidote^ with Wtf4^poison^ suppresses the deleterious and toxic effect exerted by Wtf4^poison^ through a mechanism, they argue, that requires sequestration of the toxic Wtf isoform into either a spatially confined compartment (IPOD or PAS – observed in S. cerevisiae) or directly inside the vacuole (*S. pombe*). However, it is not entirely clear whether varying mechanisms are at play regarding the suppression of toxicity in the different yeasts (is it just that spatial sequestration, and not vacuolar degradation, is sufficient for suppression of toxicity in *S. cerevisiae*?). A Western blot analysis should be performed over the course of an estradiol induction in both *S. pombe* and *S. cerevisiae* vegetative cells to confirm that the toxin is degraded in pombe but not in cerevisiae; this would also address questions regarding overall levels of the proteins. Because GFP or RFP are resistant to proteolysis, the use of the fluorescence signal as a proxy for the fusion proteins needs to be substantiated on Western blots that will distinguish the fusion protein from free GFP/RFP.

We agree that protein degradation is likely important for both *S. pombe* and *S. cerevisiae* to survive when both Wtf4 antidote and poison are expressed. To address this, we carried out Western Blots in *S. cerevisiae*. We observed bands at or exceeding the predicted sizes of the Wtf proteins, consistent with the idea that the fluorescent signal represents fusion proteins. We also observed data consistent with Wtf^poison^-GFP degradation in *S. cerevisiae* (Figure 2—figure supplement 3). These new data are discussed on in subsection “Wtf4 poison and antidote proteins assemble into aggregates individually and together in budding yeast”, and we updated discussion of our model to include that at least some of the Wtf4 proteins are likely degraded in the vacuole (subsection “Wtf4^antidote^ promotes neutralization of Wtf4^poison^ via recruitment to vacuole-associated sites”).

In addition, free mCherry and GFP do not FRET (see https://www.picoquant.com/scientific/practical-manual-for-fluorescence-microscopy) so our FRET data (Figure 2F, Figure 2—figure supplement 1F, Figure 5C) argues that we are observing fusion proteins. Similarly, we show that mEOS monomers do not FRET, but the Wtf-mEOS fusion proteins do (Figure 2—figure supplement 4C).

3) A clear demonstration that Wtf4^antidote^ physically interacts with Wtf4^poison^ would be important for understanding the mechanism of toxicity suppression. The authors use FRET to provide evidence that the proteins indeed physically interact. To further support this, can the authors biochemically demonstrate a physical interaction between these proteins, for example by co-immunoprecipitation or another method? Short of demonstrating a direct physical interaction, it would be helpful to know whether both proteins come down in a pelletable aggregate.

We attempted, but were unable to immunoprecipitate the Wtf proteins, perhaps due to their low solubility. We show in Figure 2—figure supplement 3 that the two proteins are found mostly in the pellet of our protein preparation. Our FRET and AmFRET experiments do, however, show a direct physical interaction between the Wtf proteins (Figure 2F, Figure 2—figure supplement 1F, Figure 5C, Figure 2—figure supplement 4C).

In addition, the allele-specific interactions we observed with our *wtf4^antidote^** and *wtf4^poison^** alleles (Figure 3) provides additional evidence of a sequence-dependent physical interaction between Wtf proteins.

4) The colocalization between the Wtf4 protein aggregates and the IPOD marker Rnq1-mCardinal isn't overly convincing. Can the authors quantify the number of cells that exhibit this kind of distribution?

We have repeated the analysis of Wtf4 protein localization relative to Rnq1-mCardinal, as well as the PAS marker, GFP-Atg8. We have quantified the localization patterns and present the data in Figure 5—figure supplement 1-2. These results are described in subsection “Wtf4 poison-antidote protein aggregates often localize to the IPOD and PAS in budding yeast” in the manuscript.

[Editors' note: further revisions were suggested prior to acceptance, as described below.]

The reviewers commented positively on the revised manuscript and recommended publication when you are able to address a remaining concern of one of the reviewers:"In response to concern #2 in the decision letter, authors have only performed western blots for S. cerevisiae, while blots for both S. pombe and S. cerevisiae were requested. The blots are somewhat marginal in quality and do not contain a free GFP sample to allow unambiguous identification of the free GFP that would be diagnostic of vacuolar degradation, nor does it look as if there is much signal at the expected 27 kDa region of the gel. This, combined with the prominence of anti-GFP reactive bands of unexpected sizes, suggest that the text in subsection “Wtf4^antidote^ promotes neutralization of Wtf4^poison^ via recruitment to vacuole-associated sites” should be modified to acknowledge the possibility of other degradation mechanisms." Please address this concern in full by providing the requested data.

Previously, the reviewer-flagged text read:

“In *S. pombe* cells, the Wtf4^antidote^ aggregates enter the vacuole. In *S. cerevisiae* cells, the Wtf4^antidote^ accumulates outside the vacuole in the IPOD, but could also be trafficked into the vacuole at some rate, as we observed signs of protein degradation (Figure 2—figure supplement 3).”

We have now corrected this text to better reflect the limitations of our data to read:

“In *S. pombe* cells, the Wtf4^antidote^ aggregates enter the vacuole. In *S. cerevisiae* cells, the Wtf4^antidote^ accumulates outside the vacuole in the IPOD, but could also be trafficked into the vacuole at some rate. We observed signs of protein degradation, but this could be due to vacuolar degradation or other degradation mechanisms (Figure 2—figure supplement 3).”